# Combination radiation and αPD-L1 enhance tumor control by stimulating CD8+ PD-1+ TCF-1+ T cells in the tumor-draining lymph node

Yang Shen[1,15], Erin Connolly[2,15], Meili Aiello[1], Chengjing Zhou[1], Prasanthi Chappa[1], Haorui Song [1], Patan Tippitak[1], Tarralyn Clark[1], Maria Cardenas [3], Nataliya Prokhnevska[4], Annapaola Mariniello[5,6], Isabelle De Bruyker [7], Meghana S. Pagadala[8], Vishal R. Dhere [1], Sarwish Rafiq [9], Aparna H. Kesarwala [1], Alexandre Orthwein [1], Susan N. Thomas[7,10,11], Shirley L. Zhang[12], Mohammad K. Khan[1], J. Brandon Dixon [7,10,11], Gregory B. Lesinski [9], Michael C. Lowe [13], Haydn Kissick [3], David S. Yu[1], Chrystal M. Paulos [13], Nicole C. Schmitt [14] & Zachary S. Buchwald [1] ✉

Combination radiotherapy (RT) and αPD-L1 therapy has potential to enhance local and distant (abscopal) tumor control, however, clinical results in humans have been variable. Using murine melanoma models, we found RT + αPD-L1 increases intra-tumor progenitor CD8+ PD-1+ TCF-1+ T cells. This increase depends on trafficking of the PD-1+ TCF-1+ cells from the tumor-draining lymph node (TdLN) to the tumor. RT alone promotes the expansion and differentiation of the TdLN derived PD-1+ TCF-1+ cells into TIM-3+ GZMB+ TCF-1- effector-like cells in the tumor with further enhancement after the addition of αPD-L1. In the TdLN, combination therapy enriches for a novel PD-1+ TCF-1+ TOX- LY6A+ subset with expression of a type I interferon and migratory signature. This subset is able to traffic to the tumor and differentiate into TIM-3+ TCF-1- cells. Finally, we found that ablation of the PD-1+ TCF-1+ T cell population attenuates the enhanced tumor control observed with combination RT + αPD-L1. These results suggest that abscopal response failures may be secondary to impaired stimulation of TdLN CD8+ PD-1 + TCF-1+ T cells or an inability of PD-1+ TCF-1+ cells in the TdLN to traffic to the tumor.

CD8+ T cells play a critical role in the anti-tumor immune response. However, chronic antigen exposure in cancer leads to CD8+ T cell exhaustion with upregulation of markers including PD-1, TIM-3 as well as epigenetic changes[1–3]. Blockade of PD-1 promotes CD8+ T cell expansion and reinvigoration leading to robust clinical responses in many different types of cancer[4–7]. Interestingly, CD8+ PD-1+ T cells within the tumor microenvironment are heterogenous with subsets including progenitor PD-1+ TCF-1+ T cells and PD-1+ TIM-3+ GZMB+ effector-like cells[8,9]. Following PD-1/L1 blockade, the CD8+ PD-1+ TCF-1+ T cell subset expands and differentiates into a TIM-3+ GZMB+ subset which has the capacity for tumor killing[10,11]. Approaches which enhance this expansion and differentiation process in combination with αPD-1/L1 have the potential to further improve clinical outcomes.

Radiotherapy (RT) is effective as a local treatment and is known to also have immunomodulatory effects. On occasion, tumor regression outside the radiation field occurs via immune-stimulation, a process known as the abscopal effect[12–14]. RT mediates this effect, in part, by acting as an in-situ vaccine while broadening the T cell receptor repertoire and recruiting naïve/antigen experienced T cells to the anti-tumor immune response[14–16]. Importantly, RT can improve local and distant disease control when combined with immune checkpoint blockade including αPD-1/L1 in pre-clinical studies, however, clinical trial results have been mixed[15,17–23]. Understanding the impact of RT and combination therapy on specific T cell subsets may lead to more sophisticated integration approaches for these two treatment modalities and improved clinical outcomes.

The tumor-draining lymph node (TdLN) is important for a robust RT or αPD-1/L1 stimulated immune response[24–28]. More recent studies have shown that the TdLN acts as a reservoir for PD-1+ TCF-1+ T cells[26,29,30]. This population of PD-1+ TCF-1+ T cells in the TdLN serve as developmental precursors for the intra-tumoral population, and they continuously migrate from the TdLN to the tumor under basal conditions[29]. Once in the tumor TCF-1+ cells undergo further differentiation into TIM-3+ TCF-1- subsets. This process is promoted by αPD-1/L1[26]. Finally, our earlier work suggested this TdLN reservoir of TCF-1+ cells may also be important for the RT alone stimulated immune response[25]. Together, these findings suggest that the TdLN PD-1+ TCF-1+ T cell population is important for enhanced tumor control with combination RT + αPD-L1.

Here, using murine models of melanoma, we found that RT alone and in combination with αPD-L1 promoted significant tumor infiltration and differentiation of TdLN derived CD8+ PD-1+ TCF-1+ T cells. In the TdLN, combination therapy enriched for a novel PD-1+ TCF-1+ TOX- LY6A+ subset with expression of a type I interferon and migratory signature. This subset had the capacity to migrate to the tumor and differentiate into a TCF-1- TIM-3+ GZMB+ effector-like population. Finally, ablation of the PD-1+ TCF-1+ T cells worsened tumor control following combination therapy confirming the importance of this population to the anti-tumor activity of combination RT + αPD-L1.

## Results

### Combination RT + αPD-L1 promotes an increase in intra-tumoral CD8+ PD-1+ TCF-1+ and TIM-3+ TCF-1- T cells

We previously showed that RT alone can enhance the anti-tumor immune response leading to improved tumor control in a CD8+ T cell dependent manner[25]. Here, to interrogate the impact of combination RT + αPD-L1 on local and abscopal tumor control as well as CD8+ T cell subsets, B16F10 cells expressing the lymphocytic choriomeningitis (LCMV) glycoprotein (B16F10GP), which allow for the identification of tumor-specific T cells[25], were sequentially implanted on the bilateral flanks of wt C57BL/6 mice (Fig. 1a). Sequential implantation of the flank tumors was done to model metachronous metastatic disease. Tumor 1, the initially injected tumor, was treated with 10 Gy x 1 fraction of RT (Supplementary Fig. 1a) with or without αPD-L1 starting day 10 post-implantation[25]. Mice were sacrificed 9 days after treatment initiation (day 19) for tissue analysis (Fig. 1a). Tumor 1 growth was significantly reduced with RT alone, and tumor 2 growth also exhibited a strong trend towards slowed growth (Fig. 1b, Supplementary Fig. 1b, c). In contrast, αPD-L1 alone had minimal effect on the growth of tumor 1 or tumor 2 consistent with the known resistance of this cell line to PD-1 based therapy[31,32]. Combination RT + αPD-L1, however, slowed the growth of both the irradiated tumor 1 and the unirradiated tumor 2 to a greater extent than either monotherapy (Fig. 1b, Supplementary Fig. 1b, c). We performed the same kinetic analysis in the parental B16F10 cell line and found similar robust enhancement with RT + αPD-L1 compared to monotherapy at both the primary and abscopal site (Supplementary Fig. 1d).

Next, we investigated the anti-tumor immune response and found the number of bulk CD8+ T cells in tumor 1 and tumor 2 were not significantly increased with RT or αPD-L1 alone while combination therapy demonstrated significant increases in both tumors (Supplementary Fig. 1e). We then evaluated tumor specific CD8+ PD-1+ GP33+ T cells and again found a significant increase in tumor 1 and tumor 2 following combination treatment compared to untreated or monotherapy (Fig. 1c, d). Given the importance of the PD-1+ TCF-1+ subset for the αPD-L1 response[10,11], we investigated whether this population changed in the tumor following RT + αPD-L1. We found that both the tumor specific TCF-1+ and TCF-1- TIM-3+ populations substantially increased in tumor 1 and tumor 2 after combination therapy compared to either monotherapy alone with no significant changes in their relative frequencies (Fig. 1e–g, Supplementary Fig. 1f, g). PD-1+ TIM-3+ TCF-1- cells were predominantly GZMB+ and TOX+ consistent with the previously described effector-like subset in cancer and chronic LCMV infection (Fig. 1h)[8,33]. Combination therapy also led to significant increases in both the tumor CD8+ IFN-γ+ and IFN-γ+ TNF-α+ T cells (Supplementary Fig. 1h–k). We repeated the experiment using 8 Gy × 3 fractions and again observed slowed tumor growth with combination RT + αPD-L1 at both the primary and distant (abscopal) site (Supplementary Fig. 2a–e). We confirmed these findings with another melanoma cell line, YUMM1.7 (Supplementary Fig. 2f–j).

### The TdLN supplies the tumor with CD8+ PD-1+ TCF-1+ T cells following RT + αPD-L1

Our group and others have shown that the tumor-draining lymph node (TdLN) is an important reservoir for PD-1+ TCF-1+ T cells supplying the tumor[25,26,29,30]. Tumor antigen specific cells are found in the TdLN but not the non-TdLN or other secondary lymphoid organs like the spleen (Supplementary Fig. 3a). We have previously shown that disrupting this reservoir of lymphocytes in the TdLN using fractionated radiation impaired RT alone mediated immunostimulation[25]. These data suggest that the tumor-specific PD-1+ TCF-1+ T cell reservoir in the TdLN are the source of the increase in tumor PD-1+ TCF-1+ and PD-1+ TCF-1- T cells following combination therapy as well as the enhanced tumor control. To evaluate this hypothesis, we again confirmed that the majority of the GP33+ T cells in the TdLN were TIM-3- TCF-1+ while most of the GP33+ T cells in the tumor were TIM-3+ (Fig. 2a, b). Mice were then treated with FTY720 prior to RT or αPD-L1 to prevent lymphocyte egress from the TdLN and other secondary lymphoid organs (Fig. 2c). The TdLN analyzed throughout was taken from the RT targeted side (tumor 1 side). The percentage of circulating total lymphocytes, CD4+, and CD8+ T cells in the blood decreased significantly upon FTY720 administration (Supplementary Fig. 3b–d). In both tumors, administration of FTY720 blocked the increase of total CD8+ and PD-1+ GP33+ T cells observed following combination therapy (Fig. 2d, e, Supplementary Fig. 3e–g). Examination of the subsets found the increased numbers of tumor-antigen specific PD-1+ TCF-1+ and TIM-3+ in both tumors induced by combination therapy was also abolished by FTY720 treatment (Fig. 2f–h, Supplementary Fig. 3h–j). Notably, FTY720 also attenuated the slowing of tumor 1 and tumor 2 growth by combination RT + αPD-L1 (Fig. 2i, Supplementary Fig. 3k).

In the TdLN, the frequency and number of total CD8+ T cells following RT + αPD-L1 remained unchanged with or without FTY720 (Supplementary Fig. 4a, b). In contrast, the frequency and number of CD8+ PD-1+ GP33+ T cells significantly increased with FTY720 treatment. (Supplementary Fig. 4c–e). Importantly, the number of GP33+ PD-1+ TCF-1+ T cells was significantly increased in the TdLN with FTY720 and combination therapy (Supplementary Fig. 4f–h); the TIM-3+ did not reach significance (Supplementary Fig. 4i, j). Together, these results support the hypothesis that the increase in tumor PD-1+ TCF-1+ T cells following RT + αPD-L1 depends on their egress from the TdLN.

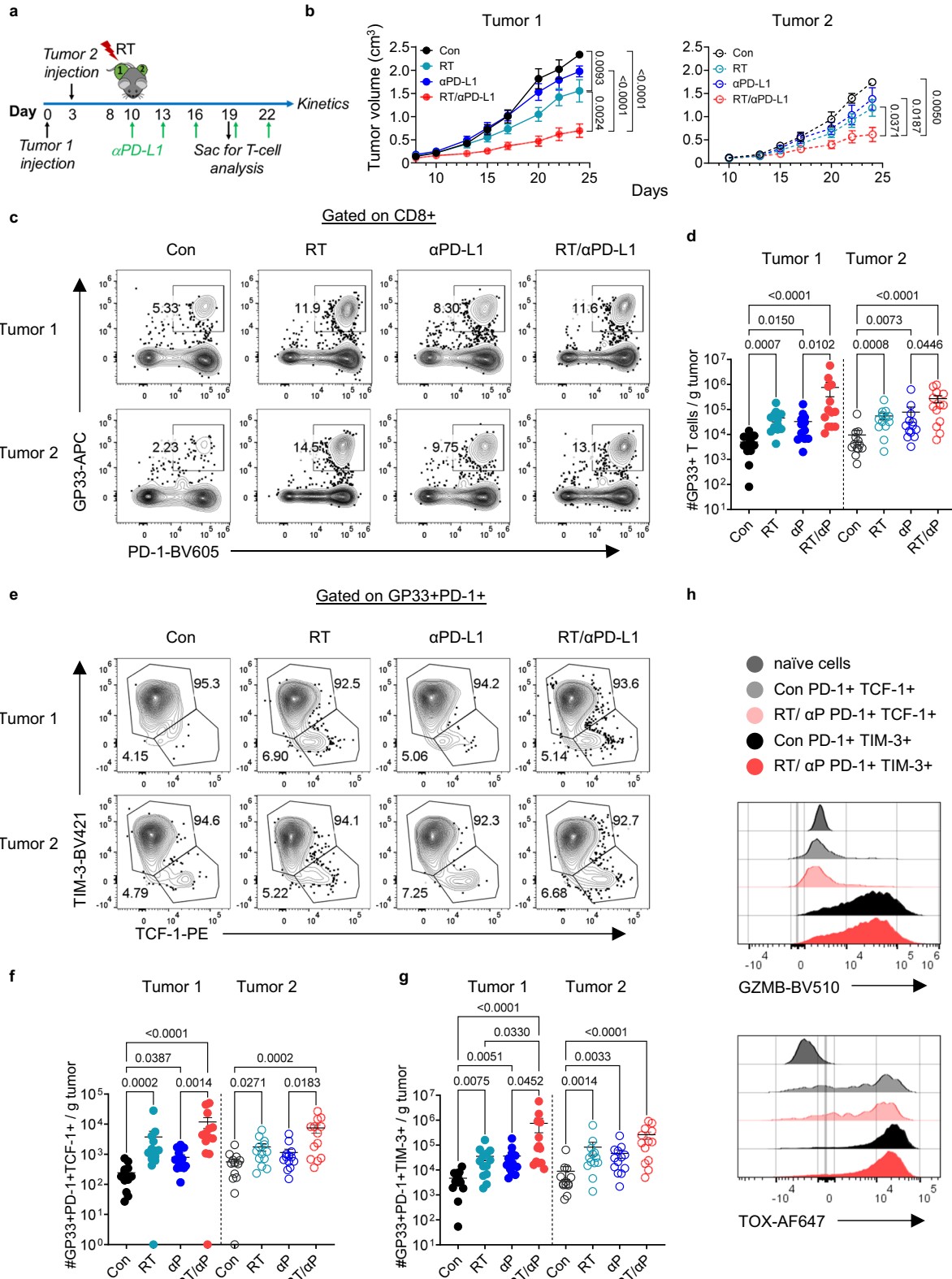

**Fig. 1 | RT + αPD-L1 promote an increase in intra-tumoral PD-1+ TCF-1+ and PD-1+ TIM-3+ CD8+ T cells. a** Experimental schema. **b** Tumor growth kinetics for the RT targeted tumor 1 and the distant (abscopal tumor) tumor 2 demonstrating enhanced control with combination therapy. Con, control; RT, radiation therapy. Data reflect 2 separate experiments combined (Con n = 8, RT n = 9, αPD-L1 n = 8, RT/αPD-L1 n = 10 total). Statistical significance calculated by two-tailed unpaired t tests. **c** Representative plots of GP33+ PD-1+ T cells gated on CD8 in tumor 1 and tumor 2 under different treatment conditions. **d** Quantitation plots for number of GP33+ T cells per gram tumor. **e** Representative plots of PD-1+ TCF-1+ and

PD-1+ TIM-3+ gated on CD8+ PD-1+ GP33+ T cells. **f** Quantitation plots for number of PD-1+ TCF-1+ T cells per gram tumor. **g** Quantitation plots for number of PD-1+ TIM-3+ T cells per gram tumor. The vast majority of GP33+ T cells in the tumor are TIM-3+ , therefore, the plots showing total GP33+ T cells (**d**) and the TIM-3+ subset are very similar. **h** Representative histogram flow plots. Data reflect 3 separate experiments combined (n = 13 total). GZMB, granzyme B. All data are presented as mean values ± SEM. Statistical significance calculated by Kruskal-Wallis test, unless otherwise noted. Source data are provided as a Source Data file.

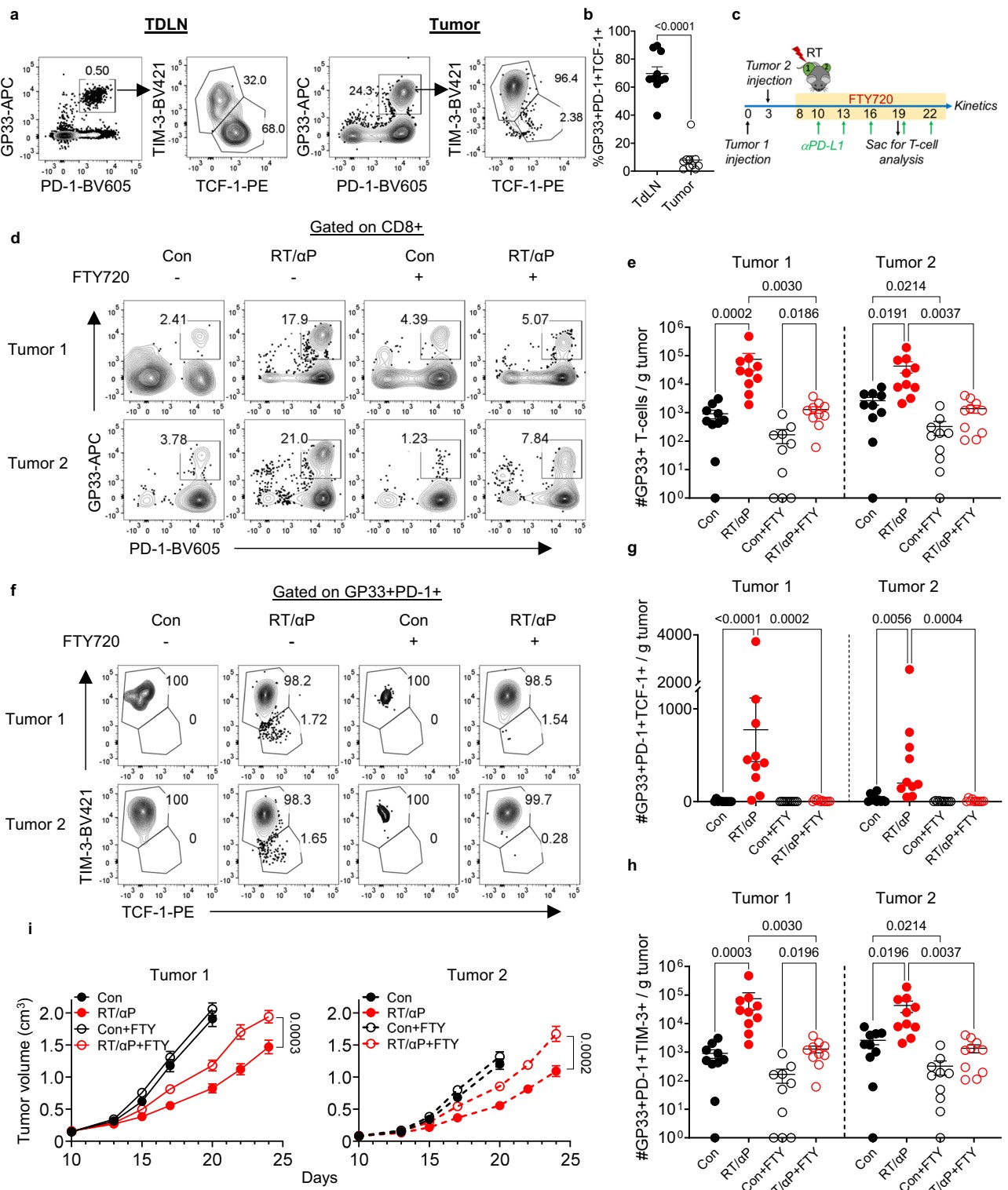

**Fig. 2 | The TdLN supplies the tumor with PD-1+ TCF-1+ CD8+ T cells following RT + αPD-L1. a** Representative plots of PD-1+ GP33+ T cells and PD-1+ TCF-1+ cells vs. TIM-3+ in the tumor and its TdLN. **b** Quantitation of the PD-1+ TCF-1+ cells frequency in the tumor and its TdLN. Combined data from 2 experiments (*n* = 10 total). **c** Experimental schema with yellow bar representing FTY720 administration in drinking water. **d** Representative plots gated on CD8 showing PD-1+ GP33+ T cells in the tumor under different treatment conditions with or without FTY720. **e** Quantitation of the number of GP33+ T cells per gram tumor. Combined data from 2 experiments (*n* = 10 total). **f** Representative plots gated on antigen specific subsets under different treatment conditions with or without FTY720.

**g** Quantitation of the number of antigen specific PD-1+ TCF-1+ T cells per gram tumor. Combined data from 2 separate experiments (*n* = 10 total per group). **h** Quantitation of the number of antigen specific PD-1+ TIM-3+ cells per gram tumor. Combined data from 2 separate experiments (*n* = 10 total per group). **i** Tumor kinetics under different treatment conditions with and without FTY720. Statistical significance calculated by two-tailed unpaired t tests. Combined data from 2 experiments (*n* = 15 total). All data are presented as mean values ± SEM. Statistical significance calculated by Kruskal-Wallis test, unless otherwise noted. Source data are provided as a Source Data file.

## RT promotes the expansion and differentiation of TdLN PD-1+ TCF-1+ T cells which is enhanced with αPD-L1

Prior data has shown that αPD-L1 monotherapy promotes the expansion and differentiation of CD8+ PD-1+ TCF-1+ T cells[10,11,26]. To determine the impact of RT alone and combination therapy on this population's differentiation, we performed a serial adoptive transfer experiment using P14 T cells. First, we sacrificed mice 14 days after a single tumor injection to confirm that adoptively transferred naïve P14 T cells would activate in the TdLN and differentiate within the TdLN and tumor into TCF-1+ and TCF-1- subsets respectively (Fig. 3a). P14s were recovered in both the TdLN and tumor, and the vast majority of the cells (99%) were PD-1+ TCF-1+ TIM-3- in the TdLN and TIM-3+ in the tumor like endogenous cells (Fig. 3b–d). We then sorted CD44+ PD-1+ TIM-3- P14s from the TdLNs of tumor bearing mice and transferred them into separate B16F10GP tumor-bearing mice (Fig. 3e, Supplementary Fig. 4k). These recipients received RT with or without αPD-L1 3 days later (Fig. 3e). We did not find any significant difference in the number of total or TCF-1+ P14s in the recipient TdLN with either monotherapy or combination (Fig. 3f, g, Supplementary Fig. 4l).

In contrast, we found a significant increase in the number of P14s in tumor 1 for RT alone (Fig. 3h, i). The frequency of PD-1+ TCF-1+ cells significantly decreased in tumor 1 with a concomitant increase in the frequency of TIM-3+ cells demonstrating that RT alone can promote both expansion and differentiation of TdLN PD-1+ TCF-1+ T cells in the RT targeted tumor (Fig. 3j–l). Importantly, combination therapy led to greater expansion of P14s in both tumor 1 and tumor 2 and enhanced differentiation of PD-1+ TCF-1+ T cells into TIM-3+ cells compared to either monotherapy alone (Fig. 3h–l).

## ScRNA-seq analysis identified multiple CD8+ PD-1+ TCF-1+ T cell subsets in the TdLN

To further interrogate the TdLN, we performed single cell RNA-seq (scRNA-seq) on sorted TdLN naïve and CD8+ PD-1+ T cells under untreated conditions. Published tumor infiltrating CD8+ T cell data from similar tumors models were also introduced into our analysis[34]. Unsupervised clustering using uniform manifold approximation and projection (UMAP) revealed substantial heterogeneity within CD8+ T cells, identifying at least six distinct subtypes across the TdLN and tumor (Fig. 4a). These clusters included: Cluster 1, naïve T cells, defined by high *Tcf7* expression and negative for activation markers including *Fos* and *Jun*; Cluster 2, stem-like-1 (T_STEM-1), defined by *Tcf7* and *Fos* expression and the absence of *Tox* expression; Cluster 3, stem-like-2 (T_STEM-2) co-expressing *Tcf7* and *Ly6a* with low *Tox* expression; Cluster 4, progenitor exhausted (T_PEX), characterized by co-expression of *Tcf7* and *Tox*; Cluster 5, effector-like and terminally differentiated (TD), defined by *Tcf7* negativity and positive *Tox* expression; and Cluster 6, cycling T cells, identified by *Mki67* expression (Fig. 4a). In the tumor, the predominant CD8+ T cell subset was Cluster 5, comprising about 50% of the CD8+ PD-1+ T cell population (Fig. 4b). In contrast, *Tcf7*-positive subsets (Clusters 2, 3, and 4) were primarily found in the TdLN, with the *Tcf7*+ *Tox*+ T_PEX population (Cluster 4) being the most abundant. The resulting clusters were validated by comparing their transcriptional signatures to known marker genes and previously published datasets, ensuring the identified subsets were biologically meaningful and consistent with established T cell populations (Supplementary Fig. 5a)[26].

To evaluate for a clonal relationship between TdLN-derived CD8+ PD-1+ T cells and CD8+ T cells within the tumor, we performed bulk-TCR sequencing on sorted polyclonal CD8+ PD-1+ T cells from paired TdLN and tumor samples from untreated mice. TCR sequencing demonstrated that up to 50% of the tumor TCR repertoire overlaps with the TdLN, suggesting that a significant fraction of antigen-experienced TdLN T cells are tumor-specific, even under untreated conditions (Fig. 4c).

A hallmark of T cell exhaustion is the sustained expression of markers such as *Tox, Ctla4, Entpd1, Pdcd1,* and *Havcr2*, many of which were enriched in the tumor-infiltrating subset (Cluster 5) (Fig. 4d, e). Notably, exhaustion markers such as *Lag3, Ctla4, and Tox* were also expressed by the T_PEX subset, distinguishing them from other TCF-1+ populations (Fig. 4e). Additionally, Cluster 5 in the tumor exhibited elevated expression of effector genes, such as *Gzmb, Ifng,* and *Klrk1*, linked to cytotoxic T cell functions (Fig. 4d, e). In contrast, TdLN PD-1+ CD8+ T cells (Clusters 2 and 3) were enriched for stemness markers (e.g., *Tcf7, Il7r, Sell, Ccr7*) and activation markers (e.g., *Jun, Fos, Cd69, Junb*) (Fig. 4d). We identified a distinct stem-like subset (Cluster 3, T_STEM-2), marked by the co-expression of an interferon-stimulated gene signature including type I interferon response genes (*Isg15, Irf7,* and *Ifitm3*), chemokine markers (*Ccr5* and *Ccrl2*), and the murine Ly6 gene complex, including *Ly6a* and the memory marker *Ly6c* (Fig. 4e)[35]. This subset exhibited a unique profile, characterized by the presence of both stemness-associated and effector genes. Notably, the expression of effector genes such as *Klrk1* and *Gzmb* distinguished T_STEM-2 from the canonical T_STEM-1 subset (Cluster 2), which lacked these markers. Given T_STEM-2 exhibits characteristics of both stemness and differentiation, this subset may play a transitional role within the CD8+ T cell response in the TdLN despite the lack of *Tox* expression like the canonical T_PEX (Fig. 4e).

Next, to characterize the broader pattern of T cell phenotypes across the TdLN and tumor, we identified two highly correlated gene modules: a stemness module in the TdLN and an exhaustion module in the tumor, consistent with prior reports (Fig. 4f, g)[26,29]. While these modules do not directly correspond to cluster-specific marker genes, they provide complementary insights, capturing dynamic transcriptional programs associated with differentiation states. These data support the idea that T_STEM-1 and Cluster 5 occupy opposite ends of the T cell differentiation spectrum.

## Combination RT + αPD-L1 enriches for a TCF-1+ TOX- subset with a type I interferon and migratory signature in the TdLN

To assess the impact of combination therapy on these different antigen experienced TCF-1+ subsets within the TdLN, we performed scRNA-seq on sorted CD8+ PD-1+ T cells from the TdLN seven days post-treatment with αPD-L1, RT alone, or combination RT + αPD-L1 therapy. A total of 38,578 cells were analyzed, with an average of 1928 cells per sample across five mice per treatment group. Unsupervised clustering of the TCF-1+ cells identified 3 major clusters (T_STEM-1, T_STEM-2, T_PEX) in the TdLN under different treatment conditions (Fig. 5a). CD8+ PD-1+ T cell subset frequencies from untreated mice were largely similar to those treated with αPD-L1 or RT monotherapy. However, combination therapy led to a notable phenotypic shift, marked by an over 10-fold increase in the frequency of the T_STEM-2 subset (Fig. 5a–c, Supplementary Fig. 5b), which was accompanied by a reduction in the frequency of both the T_STEM-1 and T_PEX population (Fig. 5a–c, Supplementary Fig. 5b). Differential abundance analysis revealed that the expansion of the T_STEM-2 subset was statistically significant (FDR < 0.05) when comparing RT + αPD-L1 to monotherapies or control (Fig. 5c).

Given these phenotypic changes, we next quantified differentially expressed genes (DEGs) in each treatment group relative to untreated controls to explore therapy-induced changes in gene expression. Combination therapy with RT and αPD-L1 resulted in a markedly higher number of upregulated DEGs in CD8+ PD-1+ T cells compared to either monotherapy alone (Combo = 178 genes; RT = 5 genes; αPD-L1 = 6 genes), with minimal overlap in upregulated genes across the three treatment groups (Supplementary Fig. 5c, d). In contrast, relatively few genes were significantly downregulated across all three treatment cohorts (Combo = 13 genes; αPD-L1 = 14 genes; RT = 13 genes) (Supplementary Fig. 5c), emphasizing the unique effect of combination therapy in driving gene upregulation.

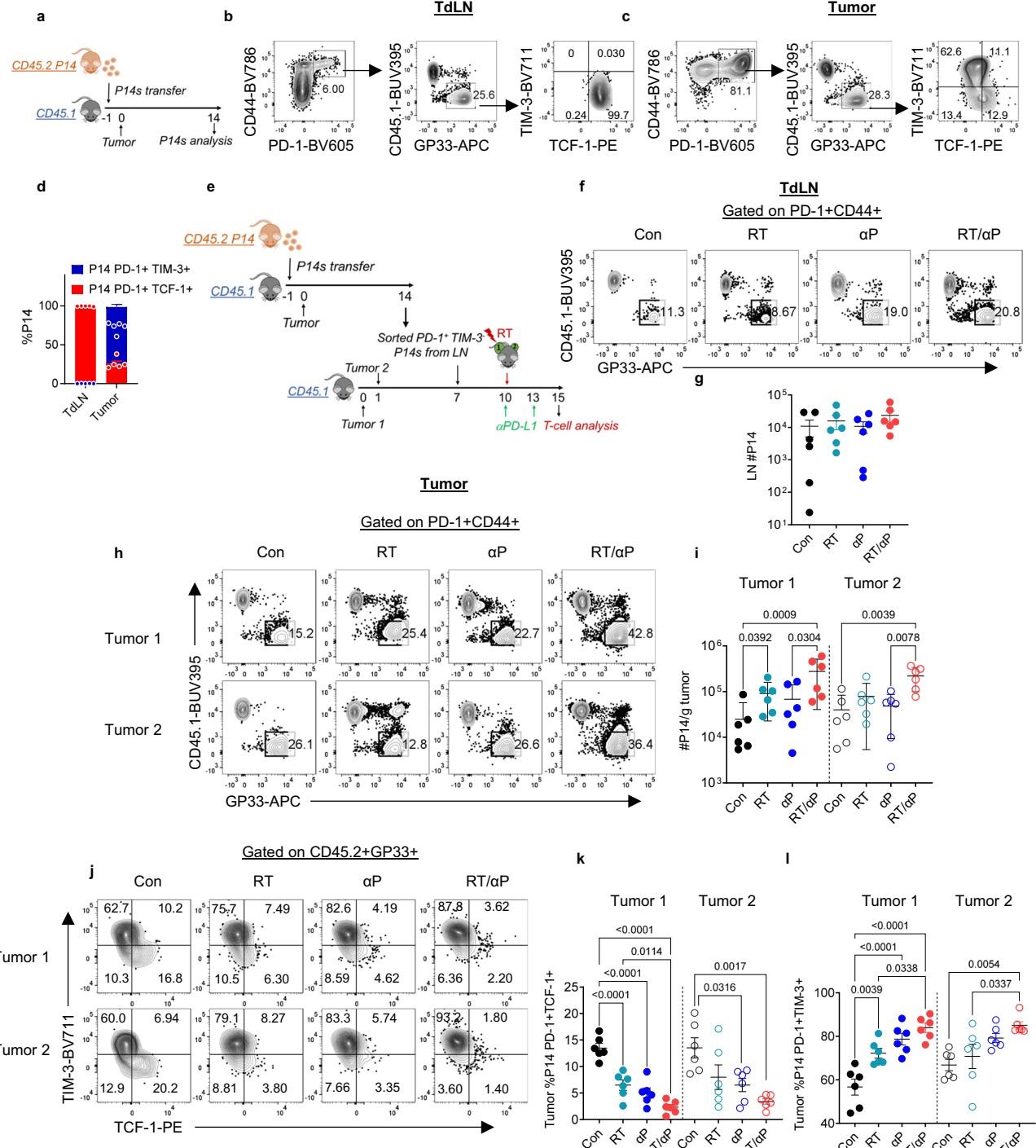

**Fig. 3 | RT promotes the expansion and differentiation of TdLN PD-1+ TCF-1+ T cells which is enhanced with αPD-L1. a** Experimental schema. **b** Representative flow plots showing the P14 T cells differentiating into PD-1+ TCF-1+ cells in the TdLN and (**c**) PD-1+ TIM-3+ in the tumor. **d** Frequency of PD-1+ TCF-1+ cells and PD-1+ TIM-3+ in the tumor versus its TdLN. **e** Experimental schema with serial adoptive transfer. **f** Representative flow plot of gating on transferred P14s in the TdLN of tumor 1 under different treatment conditions. **g** Quantitation of the number of P14s in the TdLN of tumor 1 by treatment condition. Data reflect combined data from two separate experiments (*n* = 6 total). **h** Representative flow plot of gating on transferred P14s in the tumor under different treatment conditions. **i** Quantitation of P14s per gram tumor. Data reflect combined data from two separate experiments (*n* = 6 total). Statistical significance calculated by Kruskal-Wallis test. **j** Representative flow plots of P14 T cell subsets in the tumors. **k** Frequency of PD-1+ TCF-1+ and **l** PD-1+ TIM-3+ in the tumors. Data reflect combined data from two separate experiments (*n* = 6 total). All data are presented as mean values ± SEM. Statistical significance calculated by one-way ANOVA, unless otherwise noted. Source data are provided as a Source Data file.

To better understand the functional and transcriptional relevance of the expanded T_STEM-2 subset, we examined a curated panel of key differentially expressed genes between T_STEM-2 and the other *Tcf7*-expressing subsets (Fig. 5d). Stem-like and memory-associated marker *Ly6a* was enriched in T_STEM-2 cells, consistent with their progenitor-like identity. Additionally, the migration-associated marker *Ccrl2* and the activating receptor *Klrk1*, commonly associated with NK cells, were upregulated alongside the effector molecule *Gzmb*. In contrast,

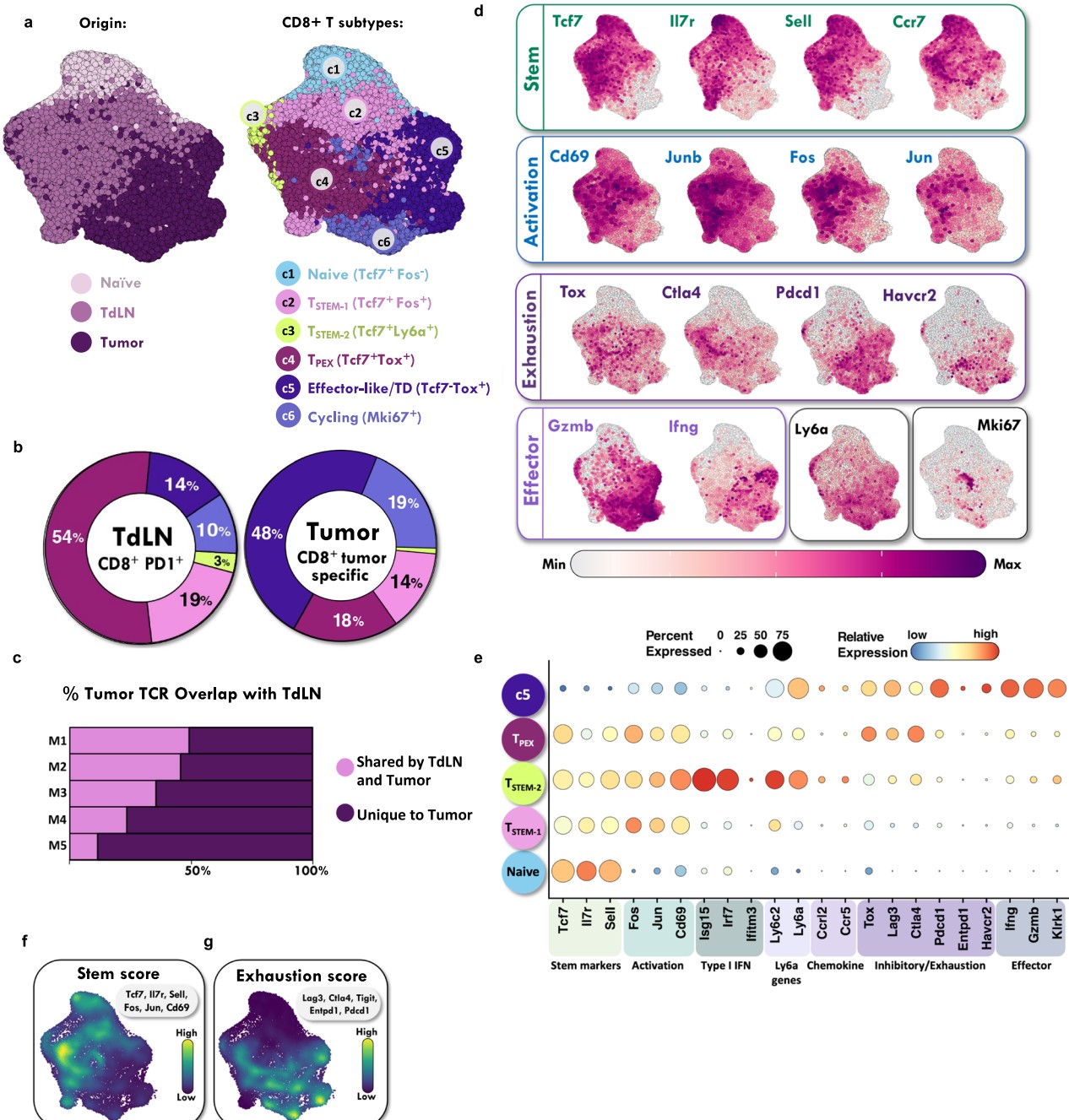

**Fig. 4 | ScRNA-seq analysis identified multiple CD8+ PD-1+ TCF-1+ T cell subsets in the TdLN. a** UMAP (Uniform Manifold Approximation and Projection) identified six major cell populations in the TdLN and tumor. **b** Quantitation of different subset frequencies in the TdLN and the tumor; with subset identities as indicated in (**a**). **c** TCR (T-cell receptor) sequencing demonstrates percentage overlap between antigen experienced polyclonal CD8+ T cell populations in the tumor and TdLN.

Each row (M) reflects a different mouse. **d** Feature plots showing expression levels of relevant markers. **e** Average expression and percent expressing various markers in the different T cell subsets. **f** Density plot of stemness module score (*Tcf7, Il7r, Sell, Fos, Jun, Cd69*). **g** Density plot of exhaustion module score (*Lag3, Ctla4, Tigit, Entpd1, Pdcd1*). Source data are provided as a Source Data file and are available on the NCBI Gene Expression Omnibus (GEO) database.

exhaustion markers such as *Tox* were predominantly expressed in the $T_{PEX}$ subset, highlighting distinct transcriptional states.

Building on these findings, we explored $T_{STEM-2}$-specific transcriptional responses across different treatment conditions. $T_{STEM-2}$ cells displayed distinct transcriptional changes with effector genes such as *Gzmb* and *Klrk1* upregulated in all treatment groups (RT + αPD-L1, RT, and αPD-L1) compared to the control, suggesting enhanced cytotoxic potential across therapeutic contexts. Exhaustion markers (*Dapl1, Ctla4, Dusp1,* and *Btla*) were markedly reduced, particularly in

the combination therapy group. Stem-related genes (*Tcf7, Il7r,* and *Sell*) remained consistently expressed. Of note, *Ly6a* genes (*Ly6c2* and *Ly6a*), migration-related genes (*Cxcr3, Ly6c2, Cxcl10, Ccrl2,* and *Icam1*), cytokine receptors (*Il18rap, Ifngr1,* and *Il18r1*), and type I interferon response genes ((*Irf7, Isg15, Ifitm3* and *Stat3*) were elevated in $T_{STEM-2}$ cells compared to other subsets under all treatment conditions; however, these pathways were further upregulated in the $T_{STEM-2}$ cells of the combination therapy group. Other subsets, including $T_{STEM-1}$ and $T_{PEX}$, exhibited less pronounced transcriptional changes in

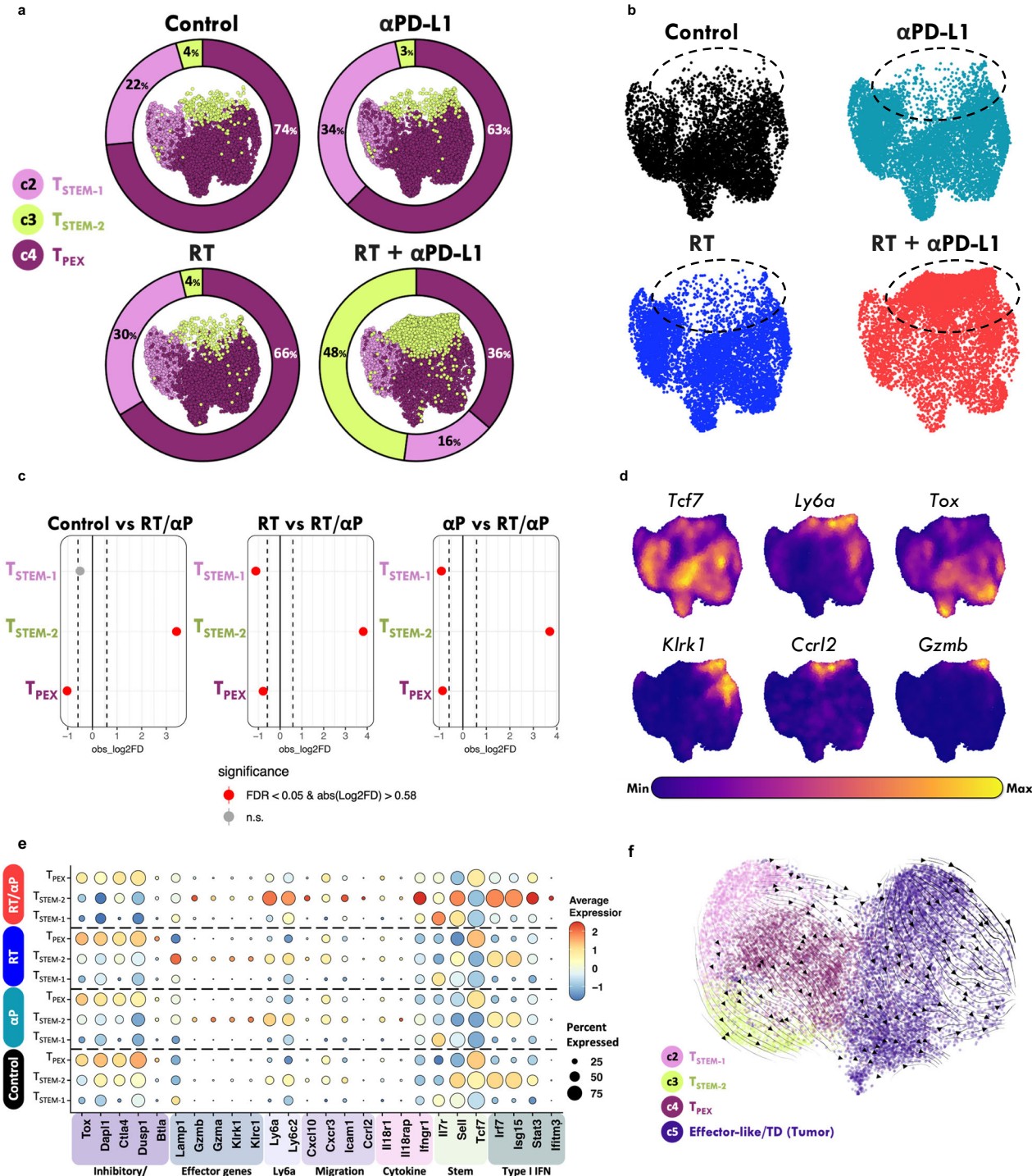

**Fig. 5 | Combination RT + αPD-L1 expands a novel TCF-1+ subset in the TdLN.**
**a** UMAP and quantitation demonstrating the major cell PD-1+ *Tcf7*-expressing T cell populations in the TdLN under different treatment conditions. **b** UMAP by treatment condition with the novel population in RT + αPD-L1 group circled.
**c** Proportions of *Tcf7*-expressing CD8+ PD-1+ T cell subtypes were compared across treatment conditions using a two-sided permutation test (1000 iterations), with empirical *P* values adjusted by the Benjamini−Hochberg method. Red dots indicate significant differences (FDR < 0.05, |log$_2$FC| > 0.58); gray dots are non-significant. Exact *p* values and 95% confidence intervals (bootstrapped, 1000 iterations) are reported. **d** Density plots showing expression levels of *Tcf7, Ly6a, Tox, Klrk1, Ccrl2,*

and *Gzmb* in CD8+ *Tcf7*-expressing PD-1+ T cells from the TdLN, with color intensity representing scaled expression levels (purple = minimum; yellow = maximum).
**e** Gene expression patterns across CD8+ PD-1+ *Tcf7*-expressing T cell subsets under different treatments (RT, αPD-L1, RT + αPD-L1) compared to controls. Dot size represents percent expression, and color indicates average expression levels for exhaustion, effector, Ly6a, migration, cytokine receptor, stem, and Type I interferon (IFN) genes. **f** RNA velocity analysis of CD8+ PD-1+ T cell subsets with arrows indicating inferred directional transitions between T$_{STEM-1}$, T$_{STEM-2}$, T$_{PEX}$, and Cluster 5 (Effector-like/TD) states. Source data are provided as a Source Data file and are available on the NCBI Gene Expression Omnibus (GEO) database.

response to treatment, emphasizing the unique impact of combination therapy on $T_{STEM-2}$ cells (Fig. 5e, Supplementary Fig. 5e, f).

Given the $T_{STEM-2}$ cell subset expression of both stemness and effector-like markers, we evaluated the differentiation trajectories of the TdLN TCF-1+ populations. RNA velocity was performed on cells under combination RT + αPD-L1. The velocity vector field, visualized on the UMAP embedding (Fig. 5f), points to distinct pathways originating from $T_{STEM-1}$ and progressing towards either $T_{STEM-2}$ or $T_{PEX}$ populations. These findings suggest the $T_{STEM-2}$ are an intermediate in a non-canonical differentiation program driven by combination RT + αPD-L1 therapy.

### $T_{STEM-2}$ from the TdLN traffic to the tumor where they undergo differentiation

Next, to investigate these scRNA-seq findings further and determine if they correlated with protein expression, we adoptively transferred tumor specific CD45.2+ P14s into CD45.1 mice, implanted a single tumor, treated with RT + αPD-L1, and then, on day 17, we co-stained the transferred cells in the TdLN and tumor. In the TdLN, again gating on PD-1+ CD44 + P14s, we observed both TCF-1+ LY6A+ and TCF-1+ LY6A- subsets (Fig. 6a). Within the TCF-1 + LY6A- population, we found a TOX+ and TOX- population which were defined as $T_{STEM-1}$ and $T_{PEX}$ respectively[22]. Within the LY6A+ population, we further identified a CD314+ subset, and consistent our scRNA-seq data, this was termed the $T_{STEM-2}$ population. We found a > 10 fold increase in the $T_{STEM-2}$ population between control and combination therapy as in the scRNA-seq data. There were also notable, but lower magnitude increases in $T_{PEX}$ and $T_{STEM-1}$ (Fig. 6b). In the tumor, a TOX+ TCF-1- subset was observed (Fig. 6c). The MFI for a selection of markers varied somewhat across these different subsets and was consistent with the scRNA-seq data (Fig. 6d). Tumor-specific P14 $T_{STEM-2}$ cells were also identified in the blood following RT + αPD-L1 (Fig. 6e), suggesting they can traffic from the TdLN to other tissues including the tumor. A similar increase in the TdLN $T_{STEM-2}$ population was observed with another tumor, YUM-MER1.7, treated with combination RT + αPD-L1 (Supplementary Fig. 6a–c), demonstrating the findings are not limited to a single model.

Since the tumor-specific $T_{STEM-2}$ are identifiable in the blood and express a migratory transcriptional signature, we performed an adoptive transfer experiment to determine whether they can infiltrate the tumor and can further differentiate into effector-like cells (Fig. 6f). P14 $T_{STEM-2}$ from the TdLN were sorted and transferred into B16F10GP single tumor-bearing mice (Fig. 6f, Supplementary Fig. 6d). 7 days later, the transferred cells were in the tumor and had downregulated TCF-1, CD62L, increased expression of PD-1 and upregulated GZMB and TIM-3 demonstrating an ability to differentiate into an effector-like subset (Fig. 6g, h).

### CD8+ PD-1+ TCF-1+ T cells are required for enhanced tumor control with combination RT + αPD-L1

Finally, given the broad impact of RT/αPD-L1 on the PD-1+ TCF-1+ subset in the TdLN, we evaluated whether this subset was required for the enhanced tumor control observed with combination therapy. We generated a knock-in mouse allowing for specific depletion of TCF-1+ T cells. A diphtheria toxin receptor (DTR) P2A eGFP gene was inserted into the 3′ untranslated region of the *Tcf7* locus using CRISPR technology (Supplementary Fig. 7a). CD45.2 Tcf7$^{DTR-eGFP}$ were then bred with P14 mice to generate CD45.2 P14 Tcf7$^{DTR-eGFP}$ (Supplementary Fig. 7a). To verify that P14 Tcf7$^{DTR-eGFP}$ (P14 DTR+ ) activate and differentiate into T cells expressing both PD-1 and TCF-1 as well as eGFP, we adoptively transferred CD45.2 P14 DTR+ cells into CD45.1 mice on day -1. B16F10GP cells were then injected on bilateral flanks on day 0. We found P14 DTR+ T cells in both tumors, TdLN and expressing PD-1 at all sites (Supplementary Fig. 7b–d). eGFP was highly expressed in the TCF-1+ subset in both tumors and TdLN, but not in the TCF-1- TIM-3+ T cells from the tumors on day 19 (Supplementary Fig. 7e, f). Of note, PD-

1+ TCF-1+ P14 DTR+ numbers in the TdLN and tumors were similar to P14 DTR- suggesting no differences in response to chronic antigenic stimulation (Supplementary Fig. 7g, h). Tumor growth kinetics for both tumor 1 and tumor 2 in P14 DTR+ and DTR- recipients were indistinguishable (Supplementary Fig. 7i).

Next, we tested whether diphtheria toxin (DT) specifically depleted the TCF-1+ T cell population. DTR- or DTR+ P14 were adoptively transferred followed by bilateral tumor inoculations and DT administration (Supplementary Fig. 8a). In both the TdLN and the bilateral tumors, DT ablated TCF-1+ cells from adoptively transferred P14 DTR+ but not from P14 DTR- littermate controls (Supplementary Fig. 8b–e). There was also a reduction in TIM-3+ TCF-1- population (Supplementary Fig. 8f) attributable to the elimination of the precursor TCF-1+ T cells. Importantly, endogenous PD-1+ TCF-1+ T cells and PD-1+ TIM-3+ were unchanged in either DTR- or DTR+ recipients in the TdLN and both flank tumors (Supplementary Fig. 8g–i).

Having validated specific depletion of TCF-1+ T cells in CD45.1 recipient mice, we explored the impact of PD-1+ TCF-1+ T cell depletion on RT + αPD-L1. To do this, we again adoptively transferred P14 DTR+ or P14 DTR- from littermate controls into CD45.1 and implanted tumors on bilateral flanks (Fig. 7a) followed by combination therapy starting on Day 12. In the TdLN, the adoptively transferred P14+ DTR- T cells were detectable, had robustly upregulated PD-1 and were nearly all TCF-1+ , while the TCF-1+ cells in P14 DTR+ recipients were all depleted (Fig. 7b–e, Supplementary Fig. 9a). Additionally, in the P14 DTR- recipient mice treated with RT + αPD-L1, we could detect the LY6A+ cells in the TdLN draining the irradiated tumor (tumor 1), and this TCF-1 expressing subset was also ablated in the P14 DTR+ recipients (Fig. 7f, g). Next, we evaluated tumor 1 and tumor 2 and found transferred P14s in both the P14 DTR+ and P14 DTR- recipients with all expressing high levels of PD-1 (Fig. 7h, i). The TCF-1+ T cell subset was again specifically depleted in only P14 DTR+ recipients (Fig. 7j, k, Supplementary Fig. 9b–d). A reduction was also observed in both tumors of the TIM-3+ subset again confirming that the PD-1+ TCF-1+ T cells are necessary for TIM-3+ TCF-1- production (Fig. 7l, Supplementary Fig. 9e). Finally, we evaluated the growth of tumors 1 and 2 and found that with specific PD-1+ TCF-1+ depletion, both local tumor control and the abscopal effect induced by combination RT + αPD-L1 were significantly reduced (Fig. 7m, Supplementary Fig. 9f). It is likely that the endogenous PD-1+ TCF-1+ are responsible for the DTR+ group still demonstrating slowed tumor kinetics compared to controls. The results demonstrate that PD-1+ TCF-1+ T cells are important for optimally enhanced tumor control observed with combination RT + αPD-L1.

## Discussion

The aim of our study was to mechanistically dissect the abscopal effect mediated by combination RT + αPD-L1 to enhance the translational impact and guide approaches to overcome treatment failure in humans. Here, using murine melanoma models, we found that combination therapy robustly stimulates PD-1+ TCF-1+ T cell migration from the TdLN and expansion/differentiation in the tumor. Within the TdLN, RT + αPD-L1 expanded a novel subset that expresses *Tcf7, Klrk1* and *Ly6a* and has both a migratory and type I interferon signature. This LY6A+ CD314+ subset can migrate to the tumor and differentiate into TIM-3+ GZMB+ TCF-1- effector-like cells. Finally, we showed that PD-1+ TCF-1+ T cells are important for the enhanced tumor control observed with combination RT + αPD-L1. These data have several biological and clinical implications.

Biologically, RT has been previously shown to promote the release of both cryptic/sequestered tumor antigen, type I interferon signaling, and damage associated molecular patterns (DAMPs) leading to enhanced APC maturation and T cell activation[12,36]. Importantly, prior studies primarily focused on the intra-tumoral T cells, largely neglecting the T cell subsets present in secondary lymphoid organs[14]. Our findings suggest that the RT induced antigen bolus and/or cytokine production

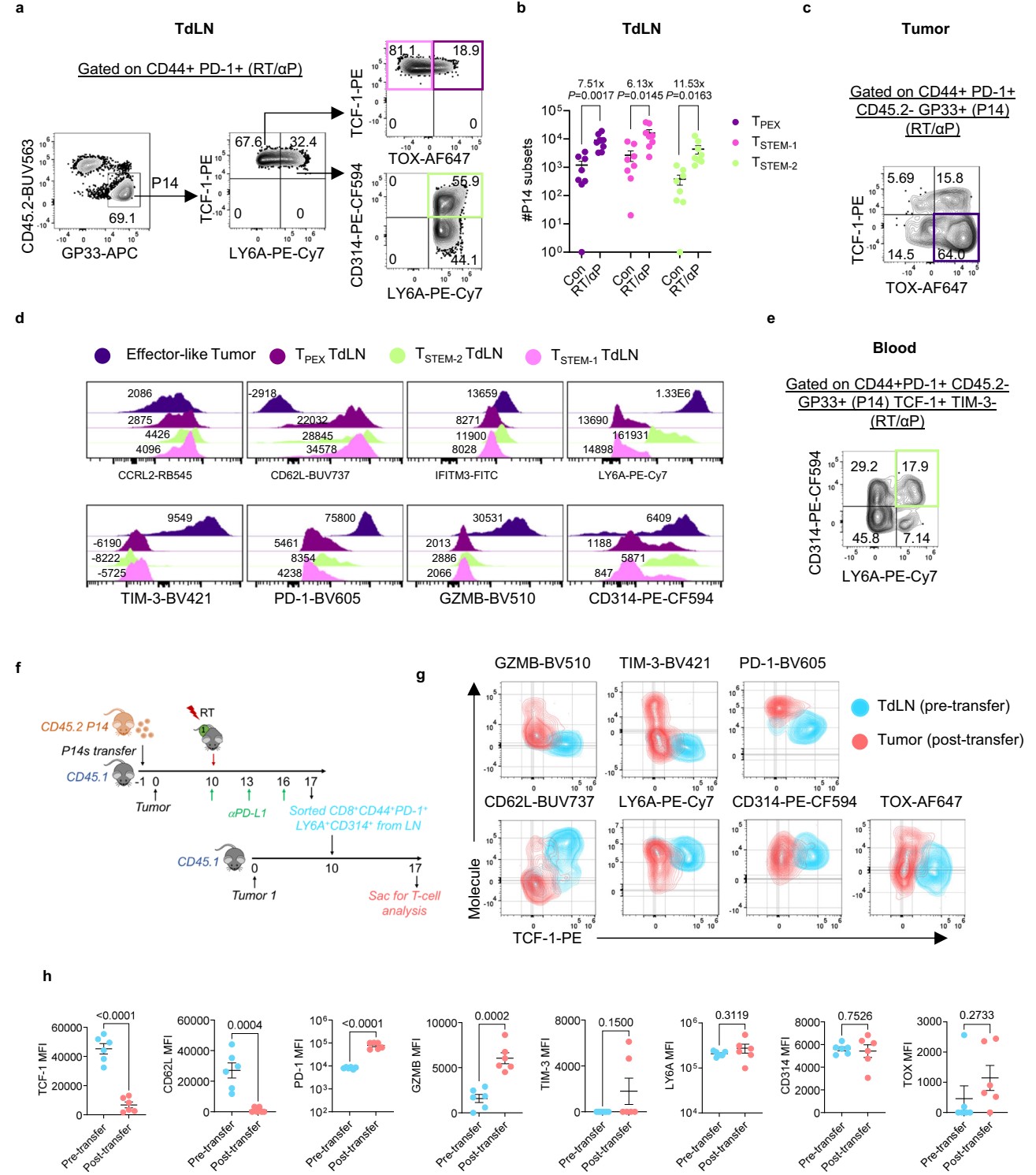

**Fig. 6 | T_STEM-2 cells differentiate into TIM-3+ T cells in the tumor.**
**a** Representative flow plots showing the gating strategy for identification of TCF-1+ subsets among transferred P14 cells in the TdLN following RT + αPD-L1.
**b** Quantification of P14 subsets in the TdLN. Data are combined from two separate experiments ($n = 8$ total). **c** Representative flow plot gating on transferred P14s in the tumor. **d** Representative histogram plots for the different subsets.
**e** Representative flow plots of P14 TCF−1+ LY6A+ CD314+ cells in the blood.

**f** Experimental schema for serial adoptive transfer. **g** Representative flow plots depicting expression of various markers on T_STEM-2 cells pre- and post-transfer. **h** Quantification of MFIs (mean fluorescence intensity) of different markers; data reflect combined data from two separate experiments ($n = 6$ total). All data are presented as mean values ± SEM. Statistical significance calculated by two-tailed unpaired t test. Source data are provided as a Source Data file.

(including type I interferons) is enough, by itself, to promote PD-1+ TCF-1+ T cell expansion and differentiation initiated in the TdLN (Fig. 3i–l)[36,37]. This is further enhanced and modified by the presence of αPD-L1. This observation is novel as, to this point, robust PD-1+ TCF-1+ T cell differentiation was thought to be almost exclusively dependent on PD-1/L1 blockade. Whether APC migration from the tumor to the TdLN or whether antigen passively drains to the node following RT is an area of active investigation and the focus of future studies.

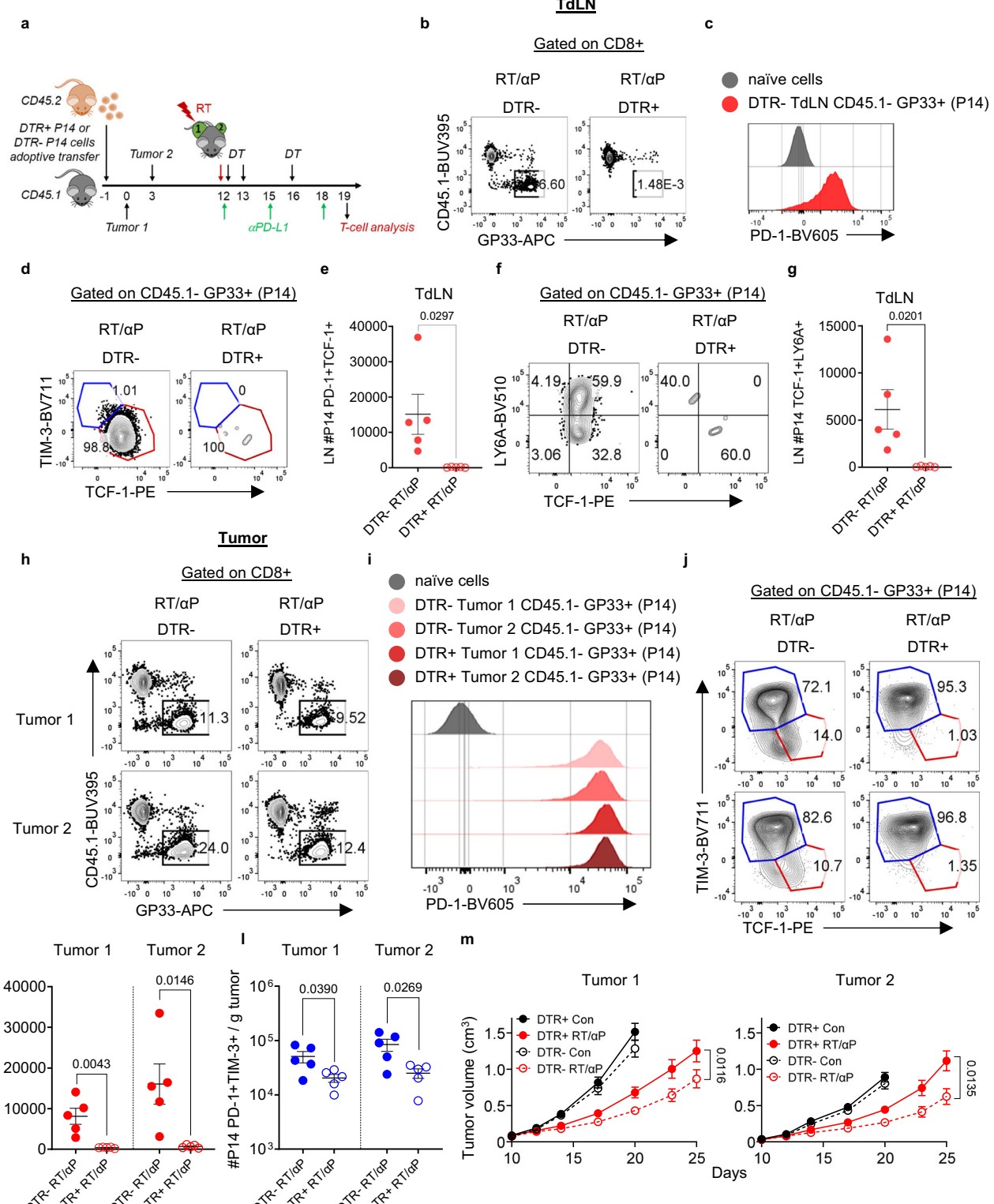

**Fig. 7 | PD-1+ TCF-1+ CD8+ T cell depletion attenuates the enhanced tumor control of RT + αPD-L1. a** Experimental schema with diphtheria toxin (DT) reflecting time points of DT administration. **b** Representative flow plots of DTR- or DTR+ P14 cells in the TdLN of tumor 1. DTR, diphtheria toxin receptor. **c** Representative histogram plot of PD-1 expression in the DTR- P14 T cells in the TdLN of tumor 1. **d** Representative flow plots of P14 subsets in the TdLN of tumor 1 for DTR- and DTR+ . **e** Quantitation of P14 PD-1+ TCF-1+ T cells in the TdLN. **f** Representative flow plots of LY6A+ TCF-1+ cells in the TdLN. **g** Quantitation of LY6A+ TCF-1+ cells in the TdLN. **h** Representative flow plots of DTR- or DTR+ P14

cells in the tumor. **i** Representative histogram plots for PD-1 expression in DTR+ and DTR- P14 cells. **j** Representative flow plots of P14 subsets in the tumors. **k** Quantitation of the number of P14 PD-1+ TCF-1+ T cells and (**l**) TIM-3+ per gram tumor. **m** Tumor growth kinetics under different treatment conditions with DTR+ or DTR- P14 cell transfer. Data shown from a representative experiment *n* = 5 per group, repeated 3 times. All data are presented as mean values ± SEM. Statistical significance calculated by two-tailed unpaired t test. Source data are provided as a Source Data file.

Another intriguing finding from our study is the identification and significant increase of the $T_{STEM-2}$ phenotype in the TdLN following combination therapy. This subset appears to differentiate from the $T_{STEM-1}$ population and given the type I interferon signature in the $T_{STEM-2}$, we speculate interferon-α/β may play a role[36–38]. Although this subset was present at very small numbers at baseline, it also exhibited a distinct transcriptional profile post-treatment. ScRNA-seq analysis revealed elevated expression of *Cxcr3*, consistent with enhanced tumor-homing capacity, alongside production of *Cxcl10*, suggesting a chemoattractant role. These attributes suggest that $T_{STEM-2}$ cells serve as central coordinators of immune recruitment, particularly for *Cxcr3*-expressing populations such as CD8+ T cells and NK cells, while simultaneously modulating the tumor microenvironment to optimize effector cell retention and sustain anti-tumor responses. This dual functionality of migration and modulation provides a mechanistic basis for the enhanced effects of combination therapy. Validation of the chemoattractant role of $T_{STEM-2}$ cells could clarify their contribution to the immune coordination underlying combination therapy.

Our findings should also be evaluated in the context of an elegant study from Hashimoto et al., which demonstrated that in chronic LCMV, combined IL-2 + αPD-L1 treatment induced a unique T cell phenotype co-expressing *Tcf7* and effector molecules[39]. In our study, following combination therapy, the $T_{STEM-2}$ subset in the TdLN demonstrated stem and modest levels of effector gene expression suggesting that there may be shared induction mechanisms between those two subsets in different treatment and model systems. Of note, in our study $T_{STEM-2}$ cells can migrate to the tumor and continue their differentiation into TIM-3+ GZMB+ cells potentially bypassing the $T_{PEX}$ intermediate state in the TdLN. Although these results are compelling, we wish to avoid overstating these observations, and it is still possible that $T_{STEM-2}$ undergo transient $T_{PEX}$ differentiation in the tumor prior to TCF-1 downregulation[22]. Future studies will evaluate this in more detail, as well as determining whether the $T_{STEM-2}$ subset may serve as superior precursor for adoptive cell therapy and whether they can generate TIM-3+ GZMB+ with more potent effector potential.

Clinically, a number of trials evaluating combination RT + checkpoint blockade have had mixed to underwhelming results[19,21,40]. Many of the clinical trials have focused on treating larger volumes with elective nodal irradiation, in particular head and neck cancer[40]. More recent data has confirmed that elective nodal irradiation or surgical nodal disruption can blunt both the local and distant radio-immunotherapy stimulated anti-tumor response[24,25,41,42]. Our findings offer a potential explanation for the observed clinical data, providing insight into the underlying mechanisms which may also impact RT with other checkpoint inhibitors including αCTLA-4[43]. These data also suggest that a neoadjuvant approach to combination therapy, especially for melanoma, where the draining nodes are not disturbed by either surgery or radiation will have the potential for greater anti-tumor immune responses. Similarly, metastatic sites of disease targeted for induction of an abscopal response must have robust nodal drainage to effectively stimulate an immune response.

Finally, future studies will investigate methods to overcome the dependency on the TdLN. As noted, many clinical scenarios have tumors which either lack robust nodal drainage or it is difficult to assess. Therefore, if a TdLN-like microenvironment can be replicated within the tumor or other secondary lymphoid organs, then this anatomical and immunologic limitation may be overcome.

## Limitations of this study

In this study, we evaluated the importance of the TdLN and PD-1+ TCF-1+ T cells for the enhanced tumor control of RT + αPD-L1 in murine melanoma tumor models. However, human data will ultimately be needed to determine the applicability of these findings to human disease. Clinical trials to evaluate the immunologic impact of neoadjuvant RT + αPD-1/L1 in melanoma are planned. Additionally, given our studies were restricted to melanoma, other cancer types need to be investigated in the future to establish the generalizability of our findings.

## Methods

### Mice

Six-week-old female C57BL/6 mice were purchased from the Jackson Laboratory. All mice were used in accordance with the Emory University Institutional Animal Care and Use Committee guidelines (protocol #: PROTO202000109). Mice were housed under the following conditions: a light cycle from 7:00 AM to 7:00 PM, a temperature of between 68 and 72 °F, and humidity ranging from 30 to 70 g/m³. Mice were sacrificed if they become sick, lethargic or had >10% weight loss prior to tumor volume defined endpoint. The B16F10 cell line was obtained from American Type Culture Collection (ATCC #CRL-6475). A B16F10 cell line expressing the glycoprotein (GP) of the LCMV Armstrong strain was generated by lentiviral transduction. Briefly, the codon- optimized GP was cloned into the bicistronic replication deficient lentiviral vector pLVX- IRES- ZsGreen1 (Takara) followed by production of lentiviral particles in 293 T cells (ATCC) and lentiviral transduction of B16F10 cells. A stable B16F10GP cell line was established by sorting B16F10 cells expressing high levels of the green fluorescent protein ZsGreen1 using a FACS AriaII (BD Biosciences) 2 weeks after transduction. The cell line was grown in Dulbecco's modified Eagle's medium (DMEM) supplemented with 10% FBS, 100 U/mL penicillin, 100 µg/mL streptomycin, and 2 mM glutamine. The cells were cultured at 37 °C with 5% $CO_2$. YUMM1.7/YUMMER1.7 were a kind gift of the Paulos laboratory. Detailed information on the medium and chemicals used is listed in the key resources table. Tcf7$^{DTR-eGFP}$ mice was created by CRISPR/Cas-mediated genome engineering (Taconic Biosciences). The gRNA to mouse Tcf7 gene (target sequence: ATGTTGGTGCTGGCTCCACTGGG), the donor vector containing "IRES-DTR-P2A-EGFP" cassette, and Cas9 mRNA were co-injected into fertilized mouse eggs to generate targeted knock-in offspring. F0 founder animals were identified by PCR followed by sequence analysis, which were bred to wildtype mice to test germline transmission and F1 animal generation. These mice were then bred to P14 mice to generate P14 Tcf7$^{DTR-eGFP}$. PCR Primers 1: F: 5′-ACTGTGGATT-CACCCTCTGTTTAC-3′, R: 5′-ATCTTCATCACCTTAAGAGGACCC-3′. Product size: 2467 bp Wildtype allele: 469 bp. PCR Primers 2: F: 5′-CGAAGAGAAAGTGAAGTTGGGCA-3′, R: 5′-AGCTTGCCGTAGGTGG-CATC-3′. Product size: 231 bp. Homozygous: two bands with 231 and 2467 bp. Heterozygous: three bands with 231 bp, 469 bp and 2467 bp. WT: one band with 469 bp.

### Tumor irradiation

$5 \times 10^5$ B16F10GP cells were injected into the right flanks on day 0 and left flanks on day 3. After the tumor was palpable (10–12 days), the right tumors were irradiated using Small Animal Radiation Therapy (SmART +) system by Precision. During radiation, mice were anesthetized with an isoflurane-based anesthesia system. The radiation dose was 10 Gy x 1 fraction or 8 Gy x 3 fractions. The treatment protocol was planned by SmART ATP – Advanced Treatment Planning. Tumor sizes were assessed using calipers. Tumor volume was calculated according to the formula length × width × depth × 0.52. For FTY720 experiment, FTY720 was provided in the drinking water (2 µg/mL) 2 days prior to RT. FTY720 treatment was continued throughout the entire experimental course. For αPD-L1 treatment, it was administered i.p. at a dose of 200 µg per mouse. For T cell analysis, mice were sacrificed on day 19 when tumor, spleen, blood, and TdLN were harvested. Mice were monitored and euthanized in accordance with the Emory University Institutional Animal Care and Use Committee tumor burden scoring guidelines. Any tumor exceeding 20 mm in length or 2000 mm³ in volume will result in euthanasia.

## Adoptive T cell transfer

P14 cells were obtained from the spleen of P14 mice. C57BL/6 mice (CD45.1) underwent retro-orbital injection with $2.5 \times 10^5$ P14 cells one day prior to B16F10GP tumor cell implantation. P14 DTR± were used for stem-like T cell depletion experiments. For the depletion of DTR expressing cells, Diphtheria Toxin (DT) was injected i.p. 3 times at a dose of 50 mg/kg of body weight.

## Flow cytometry

Flow cytometric analysis was performed on a BD FACSymphony A3 or Cytek Aurora. Direct ex vivo staining and intracellular cytokine staining were performed with fluorochrome- conjugated antibodies. Tumor, TdLNs, blood, and spleen were harvested. Tumors were digested in Collagenase IV (300 units/mL) for 60 min in a shaker at 37 °C. TdLNs, spleen and digested tumor tissue were washed through a 70 μm filter using wash buffer (RPMI + 2% FBS) to produce a single-cell suspension. Spleen samples were ACK lysed and resuspended in FACS buffer (PBS + 2% FBS + EDTA). Tumor and blood samples underwent an additional step using lymphocyte separation medium before staining. Tissues were stained with antibodies. The list of antibodies and assays used is provided in the key resources table. To detect tumor- specific CD8+ T cells, MHC-I tetramers were prepared (The NIH Tetramer Facility). For intracellular detection of transcription factors such as T-cell factor-1 (TCF-1), cells were surface stained for 30 min, fixed and permeabilized using the Foxp3 Fixation/Permeabilization Kit according to manufacturer's instructions (eBioscience), followed by intracellular staining for 30 min. All staining was performed in a 96 well plate. Splenocytes were used for single color controls. FACS data were analyzed with FlowJo (V10.8) software.

## Cell sorting

For single-cell RNA sequencing, CD8+ PD-1+ CD44+ cells from the TdLNs were flow sorted on a FACSAria (BD) flow cytometer. Individual mice samples were hashed (BioLegend) and pooled for sequencing. CD8+ CD44- CD62L+ naïve cells from the TdLN of untreated control mice were also sorted and pooled to provide controls for the analysis. For TCR sequencing, CD8+ PD-1+ CD44+ cells from tumors were flow sorted on a FACSAria (BD) flow cytometer. For TCF-1+ P14s transfer, CD8+ PD-1+ CD44+ TIM-3- cells from TdLNs were flow sorted on a FACSAria (BD) flow cytometer. For $T_{STEM-2}$ cell transfer, CD8+ PD-1+ CD44+ LY6A+ CD314+ cells from TdLNs were flow sorted on a FACSAria (BD) flow cytometer.

## Single-cell RNA sequencing

Single-cell RNA sequencing was performed by 10x Genomics Chromium Controller.

Pre-processing of single cell RNA-seq data: The Cell Ranger Single Cell Software Suite (version 5.0.1) by 10x Genomics was used to perform de-multiplexing, barcode processing, and single-cell 3′ gene counting. Reads from each pool were then aligned to the mm10-2020 mouse genome (2020 release). The count data was processed and analyzed in R (version 4.2.1) as described below. To deconvolute the cells belonging to each sample we used the Seurat package (version 4.1.1) in R. The outputs derived from CellRanger were used to create two separate objects (one with the transcriptome alignment and one with the antibody plus hashtags (HTO) alignment). Initial objects were created using the function "Read10X". We filtered both objects based on the cell barcode to keep only cells which were identified in both the transcriptome and in the antibody alignments. After this cell filtering, we used the function "CreateSeuratObject" to create a transcriptome-based Seurat object. The antibody derived data was filtered to maintain only the hashtag counts; later it was appended as a specific assay using the "CreateAssayObject" function. For cell demultiplexing we used the function "HTODemux" with default parameters in order to maximize the number of singlets detected. Individual single cells were finally filtered based on

their assigned "HTO_classification.global"= "Singlet". Antibody reads were then normalized using the Seurat function "NormalizeData" with the parameters "normalization.method" = "CLR" and "margin"="2", to indicate a normalization across cells.

Quality control of the scRNA-seq: Low quality cells with a high percentage of mitochondrial gene counts (> -10%) and with <500 measured genes were excluded. To mitigate potential doublet inclusion, cells with UMI count above 40,000 and detected genes above 5000 were removed. A total of 20 samples were sequenced and 38,578 single cells (Untreated, 9239 cells; RT, 8932 cells; αPD-L1, 10,502 cells; RT + αPD-L1, 9905 cells) were kept for subsequent analyses. In addition, the Miller et al. and Huang et al. single-cell datasets were imported without modification as validation sets[26,34]. After filtering, data in each cell was log normalized using Seurat's 'NormalizeData' function (method = 'LogNormalize', scale.factor = 10,000), the 2000 most variable genes were identified, and the 'ScaleData' function was used to scale and center the gene expression matrix after regressing out the heterogeneity associated with cell cycle and mitochondrial contamination. For each dataset, the number of principal components used for neighborhood graph construction and dimensional reduction was set at 20. Uniform Manifold Approximation and Projection (UMAP) visualization indicated cells from different samples were well mixed into the shared space[44].

Annotation of cell clusters: To identify cell subsets, we utilized publicly available single-cell RNA sequencing datasets with comprehensive cell type annotations, specifically those from Huang et al. and Miller et al.[26,34]. Unbiased cluster identification was conducted using the Leiden algorithm, a graph-based method for community detection that optimizes modularity. This method was selected over gene enrichment score cutoffs to ensure objective cluster identification. Identified clusters were validated against known cell type markers, and differential expression analysis using the 'FindAllMarkers' function in Seurat (test.use = 'wilcox', min.pct = 0.1, logfc.threshold = 0.5) was performed to confirm the accuracy and consistency of the clusters with established cellular profiles. This combination of approaches reinforced the reliability of cell subset identification and analysis.

scProportion Analysis: To quantify the proportions of CD8+ PD1+ T cell subsets from TdLN across treatment conditions, we employed the scProportionTest package[45] which utilizes permutation-based statistical tests to detect significant differences in cell abundance. Proportion data were calculated based on the relative frequency of each subset. Results were visualized using ggplot2, illustrating the relative changes in their proportions across treatment conditions

Calculation of Gene Signature Density Plots: Gene signature density plots were generated to visualize the distribution of specific cell state signatures across cell populations. Gene signatures were curated from established literature and enrichment scores were calculated for each cell using the R package UCell[46] which applies a rank-based method to estimate the expression of gene sets within single cells. To create smoothed density plots, the Nebulosa package[47] was used, employing kernel density estimation to provide clear visualization of the distribution of signature scores across the cell populations. This approach allowed for the comprehensive assessment of various gene signature distributions in the analyzed dataset.

RNA velocity analysis: We performed RNA-velocity analysis on the TdLN dataset using velocyto (v0.17) and scvelo (v0.2.3)[40]. BAM files as generated by the BD Rhapsody WTA analysis pipeline were preprocessed with samtools to make them compatible with velocyto. Loom files generated by velocyto were loaded into scvelo to estimate and visualize RNA velocities according to the scvelo tutorial. Partition-based graph abstraction (PAGA)[41] was computed based on the RNA velocity graph, using CD8+ PD-1+ T cell subclusters as grouping variable and the option minium_spanning_tree=False. The result was visualized as a graph showing the transition confidences as directed edges. The tumor cells were derived from the Miller et al. dataset[33].

## TCR sequencing and analysis

Genomic DNA (gDNA) was extracted from sorted CD8+ PD-1+ T cells using AllPrep DNA/RNA Micro Kit (QIAGEN) according to the manufacturer's instructions. The isolated gDNA was sent to Adaptive Biotechnologies (Seattle, WA, USA) for TCR sequencing by immunoSEQ assays. In the analysis, the percentage of unique TCRs in the tumor which were overlapping and non-overlapping with the TdLN TCRs were calculated.

## Statistical analysis

All experiments were analyzed using Prism 9 (GraphPad Software). Summary graphs show means ± SEM. Statistical significance was determined as described in the figure legend.

## Reporting summary

Further information on research design is available in the Nature Portfolio Reporting Summary linked to this article.

## Data availability

Source data are provided as a Source Data file. scRNA-seq data are available in the NCBI Gene Expression Omnibus (GEO) database under the accession number GSE256178. TCR-seq data are available in the NCBI Gene Expression Omnibus (GEO) database under the accession number GSE291836. Source data are provided with this paper.

## Code availability

Custom code for scRNA-seq data analysis is available from the corresponding author on reasonable request.

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

## Acknowledgements

This work was supported by funding from ASTRO-MRA #816282, NCI 5K08CA270401 and 5U54CA274513 to Z.S.B. We would like to thank the Emory Flow Cytometry Core and the Cancer Animal Models Shared Resource for their support and services.

## Author contributions

Y.S., E.C. and Z.S.B. devised the concept. Y.S., E.C. and Z.S.B. designed experiments. Y.S., E.C., M.A., C.Z., P.C., H.S., P.T., T.C. and I.D.B. performed experiments. Y.S., E.C. and Z.S.B. performed data analysis. D.S.Y., M.C., N.P., A.M., M.S.P., V.R.D., S.R., A.H.K., A.O., S.N.T., S.L.Z., M.K.K., J.B.D., G.B.L., M.C.L., H.K., C.M.P. and N.C.S. contributed by providing feedback and their expertise. Y.S., E.C., and Z.S.B. wrote the manuscript with input from other authors. All authors reviewed the manuscript.

## Competing interests

N.C.S. has a consulting role at Checkpoint Surgical, Sensorion, and Synergy Research, Inc, is a member of the advisory board of Regeneron, receives book royalties from Plural Publishing, and has received funding from Astex Pharmaceuticals. G.B.L. has received research funding through a sponsored research agreement between Emory University and Merck and Co., Bristol-Myers Squibb, Boehringer-Ingelheim, and Vaccinex. The remaining authors declare no competing interests.

## Additional information

[1]Department of Radiation Oncology and Winship Cancer Institute, Emory University, Atlanta, GA, USA. [2]Bioinformatics Graduate Program, Georgia Institute of Technology, Atlanta, GA, USA. [3]Department of Urology and Winship Cancer Institute, Emory University, Atlanta, GA, USA. [4]Marc and Jennifer Lipschultz Precision Immunology Institute, Icahn School of Medicine at Mount Sinai (ICMMS), New York City, NY, USA. [5]Department of Pathology and Immunology, University of Geneva, Geneva, Switzerland. [6]Department of Microbiology and Immunology, Emory University School of Medicine, Atlanta, GA, USA. [7]Wallace H. Coulter Department of Biomedical Engineering, Georgia Institute of Technology and Emory University, Atlanta, GA, USA. [8]Medical Scientist Training Program, University of California San Diego, La Jolla, CA, USA. [9]Department of Hematology and Medical Oncology and Winship Cancer Institute, Emory University, Atlanta, GA, USA. [10]George W. Woodruff School of Mechanical Engineering, Georgia Institute of Technology, Atlanta, GA, USA. [11]Parker H. Petit Institute for Bioengineering and Bioscience, Georgia Institute of Technology, Atlanta, GA, USA. [12]Department of Cell Biology, Emory University School of Medicine, Atlanta, GA, USA. [13]Department of Surgery and Winship Cancer Institute of Emory University, Atlanta, GA, USA. [14]Department of Otolaryngology - Head and Neck Surgery and Winship Cancer Institute, Emory University, Atlanta, GA, USA. [15]These authors contributed equally: Yang Shen, Erin Connolly. ✉e-mail: zbuchwa@emory.edu

