## [Transparent Peer Review file · Nature Communications]

Combination radiation and α PD-L1 enhance tumor control via stimulation of a CD8⁺ PD-1⁺ TCF-1⁺ T cell population in the tumor-draining lymph node

Corresponding Author: Dr Zachary Buchwald

Version 0:

Reviewer comments:

Reviewer #1

(Remarks to the Author)

In this manuscript, Buchwald and coworkers seek to identify the mechanism that underlies the synergistic effect of radiotherapy (RT) and anti-PD-L1 immunotherapy using B16F10 melanoma cells expressing the LCMV neoantigen gp33. Using a combination of gp33-tetramers, P14 TCR transgenic mice, scRNA sequencing analysis, and a new TCF1-DTR model, the authors demonstrate an increase in two stem-like subsets (TCF1⁺Fos⁺ (Tstem) population and a TCF1⁺Tox⁺ precursor exhausted (Tpex) population) and a new TCF1⁺Ly6A⁺ effector stem (Teff stem) intermediate in the tumor draining LN. In response to radiation therapy, the LN stem-like population does not change in number; however, they appear increased in the tumor. These data are interpreted to conclude that RT is sufficient to promote stem like T cell expansion, which the authors highlight as a novel finding in their study (lines 374-8). But this conclusion should be properly tempered because it does not distinguish between the ability of RT to induce Tstem populations through 1) priming of naive T cells, which is an expected effect of RT vs. 2) expansion of pre-existing Tstem cells, which occurs following PD-1 blockade but has not been reported as a direct effect of RT. The former mechanism seems predictable, while the latter if substantiated with new data would be novel.

This study claims to have identified a novel differentiation intermediate, a T effector stem (Teff stem) subset, marked by TCF1 and Ly6A expression that appears in the tumor draining LN but not in the tumor following RT and anti-PD-L1 combined therapies. Teff stem cells were identified by scRNAseq analysis. Their intermediate state was determined by Monocle pseudotime trajectory analysis, which inferred that Tstem cells differentiate into two distinct fates, Tpex and Teff stem. A bifurcated Tstem differentiation pathway and the identification of a new Teff stem intermediate are potentially novel findings. However, pseudotime trajectory analysis is highly biased because the user chooses the root/starting point of the trajectory and inferences are made based on progressive changes in overall gene expression. RNA velocity is a far better trajectory method that is unbiased and does not require user identification of root states. Instead, it uses ratios of spliced and unspliced transcripts as an indication of active and past gene transcription. The authors need to reanalyze their trajectory analysis with RNA velocity to compare differentiation trajectories. The current pseudotime trajectory is also confusing since it predicts that Tex cells only arise from Tpex. Yet, the authors propose that with combo therapy, Teff stem cells in the LN give rise to Tex in the tumor.

This manuscript proposes the following differentiation pathway: Tstem cells give rise to Teff stem cells in the draining LN in response to combo therapy followed by Teff stem cell migration to the tumor where it expands and differentiates into Tex cells. The fate of the well-established Tpex subset in this progression are not discussed and need to be examined since the field considers Tpex cells in the LN to be the primary population responding to PD-1 blockade. The proposed differentiation pathway is also not adequately linked. If Teff stem cells are trafficking from LN to tumor, then they should be observable in the blood. If Teff stem cells that infiltrate the tumor give rise to Tex cells then there should be a progression from TCF1⁺Ly6A⁺ through intermediates to TCF1⁺Tim3⁺ cells. This should be demonstrated by flow cytometry and by adoptive transfer of intermediates to temporally link downstream states.

The overall therapeutic effect of single and even combination therapy is rather modest. This is perhaps because B16F10 and YUMM1.7 cell lines have relatively low immunogenicity and poor responsiveness to anti-PD-1 therapy. PD-1 responsive tumors (MC38, YUMMER1.7) should be examined in combination with RT to determine whether the Teff stem

intermediate is induced as a general mechanism of RT and PD-1 combination therapy.

Additional specific concerns:

1. The combination treatment appears to have an additive rather than synergistic effect. If authors wish to conclude otherwise, they should mathematically demonstrate synergism.
2. The scRNA- and TCR-seq data presented in Fig 3 lacks proper explanation in the text, figure legend and methods.
 - Fig 3A. Which specific datasets were used to determine subset identities? Was a minimum gene enrichment score cutoff imposed to give the authors confidence that each subset is accurately defined? They should use a non-biased method for cluster identification instead.
 - Fig 3B. What does M1 - M5 mean? Are these sorted cells from non-treated, RT, and/or aPD-L1 treated mice? There is no comparison to a non-tumor draining LN.
 - Fig 3C. The proportions of all 5 T cell subsets in the tumor and draining lymph node need to be verified by flow cytometry. Addition of gp33 tetramers to identify tumor-specific T cells and preferably also a non-model antigen such as Trp2 should be used to determine whether these subsets represent tumor-specific CD8 T cell populations.
 - Fig 3F. Monocle 3 pseudotime analysis was conducted to infer single cell trajectories. However, this analysis is biased and flawed given that the authors selected the TCF^{high} cells in the draining lymph node as the root/starting population for trajectory analysis. Instead, RNA velocity/scVelo analysis, which infers population transitions based on active RNA splicing of actively transcribed genes should be conducted for trajectory inference.
 - Fig 3G/H. Tstem and T exhausted signature density plots are depicted but how they were calculated is not described.
3. Fig 4A-B: As above for 3B, the proportions of all 5 T cell subsets in the tumor and draining lymph node need to be verified by flow cytometry with MHC tetramers for all 4 groups if the authors wish to highlight this population as the key subset involved in responsiveness.
 - Fig 4D-H: Text and legend are not clear as to whether data are from tumor or tumor draining LN.
 - Fig 4F: transfer of 2.5 million P14 TCR tg cells is excessive and unlikely to represent native responses to therapy.
 - Fig 4I: pseudotime analysis should be replaced with RNA velocity to avoid bias.
 - Fig S4B: y axis does not make sense. Does this represent a p-value or enrichment score? S4D: signature density plots are depicted but how they were calculated is not described.
4. While generation and use of TCF1-DTR mice to selectively deplete Tstem cells in the LN and tumor are commendable, it is predictable that TCF1⁺ cells are required for responsiveness to combined treatment.
5. gp33-tetramer staining host cells would serve as a better control in Figs 6 and 7 than the bulk T cell population.
6. Are Fig 2E and 2H the same graph?? The last combination column is different, but the other 7 columns are identical.

Reviewer #2

(Remarks to the Author)

Yang Shen et al. Radiation and anti-PD-L1 synergize by stimulating a stem-like T cell population in the tumor-draining lymph node.

Shen et al. investigated the mechanism underlying the synergism of radiotherapy and anti-PD-L1 in local and distant (abscopal) tumor control in mouse melanoma models. The study focuses on a CD8⁺ stem-like subset in the tumor-draining lymph nodes (TDLN). The main claim of the study is a novel differentiation program in TDLN stem-like cells which leads to the expansion of these cells into effector cells in tumor tissue and which is responsible for the observed synergism of RT and anti-PD-L1. The question is interesting, the methods are modern, and there are interesting observations. However, the conclusions seem to me to be partly over-interpreted and the data and conclusions are not consistently clearly described in the text.

Major points:

1. Line 51 (Abstract): "RT + anti-PD-L1 induces a novel differentiation program in the TDLN population which leads to their expansion and differentiation into effector cells in the tumor". With the methods used and data presented, can the authors be sure that this is indeed a new differentiation program? As markers for more differentiated non-stem-like cells PD1⁺TIM3⁺TCF- were used throughout the paper which are the markers typically used for (effector-like) non-terminally and terminally exhausted, but not normal effector, T cells.
2. Line 52: "effector cells". Do the authors mean exhausted T cells?
3. Line 69-71: describe the subsets of TILs as: "progenitor stem-like CD8⁺ T cells and terminally differentiated effector like cells (TE)". Not all non-stem-like cells in the TME are terminally differentiated. Therefore, this should be described better.
4. Line 91: „Huang et al. demonstrated that the TdLN stem-like population is an important mediator of the anti-PD-1/L1

response.” “our earlier work suggested...” The authors cite Huang et al. (ref. 22), but in this reference a differentiation pathway of anti-PD-1/PDL-1-responsive T cells (emphasizing the importance of TSM bona fide memory T cells rather than TPEX cells) is reported which is different from other prominent papers in the field. These different views do not become clear by reading this part of the introduction.

5. Line 118/119: the authors indicate growth of tumor 2 was significantly reduced after RT monotherapy; however, there is no indication of statistical significance between control and RT-treated mice for the abscopal tumor (Fig. 1B).

6. Line 132-134: here, the markers of the two populations mentioned (stem-like and TE) should be given in the text. According to the figures, the TE = terminally differentiated subset was identified as TCF1-Tim3+. However, the TCF1-Tim3+ population has been described by others to also contain non-terminally differentiated exhausted cells. Throughout the MS, the authors only consider stem-like and terminally exhausted T cells. Why?

7. Line 151: again, it is stated that most of the tumor-specific TILs in the tumor are terminally differentiated without measuring non-terminally differentiated cells (which are also included in the TCF-TIM3+ population).

8. Line 151 and further, and Fig. 2 A, B and further down: here and in other figures, corresponding legends and throughout the text it is unclear in which tumor (primary or secondary) the T cells were analyzed.

9. Line 178-181: The authors state they found populations with signatures similar to ref 22 (Huang et al.). Line 186-190: “Both previously described...subtypes and new groups”. It is not fully clear which subtype was previously known. Do the authors refer to the Tcf7+ Fos+ as previously known? This should be clarified.

10. Line 190: Tcf7 and GzmB are given as typical markers for the TEFF-STEM population. Why was GzmB chosen? In Fig. 3E, other markers than GzmB are much more strongly expressed in this population.

11. Fig. 2, 3, and 4: cells from TdLNs were analyzed. Were these lymph nodes draining the irradiated or the non-irradiated tumor?

12. Line 235-244: Why was Ly6a chosen and what type of molecule is it? By FACS (Fig. 4D, F) the Ly6a+ population seems to be much smaller than in the RNAseq data. Please explain.

13. Fig 4E and line 244-246 states that the Teff/stem population has less Tcf than TSTEM cells but more than TEX. Do TEX cells express Tcf?

14. Fig. 4I and line 273-276: Fig. 4I suggests that TSTEM cells are more immature than TPEX and TEFF-STEM cells. In Fig. 3, TSTEM cells represent the TCF+ Fos+ population. Is the TSTEM population in Fig. 4I the same as the TSTEM population in Fig. 3A or is TSTEM here used as a general term for stem-like cells?

15. Line 297-98: what do you mean by “the TdLN stem-like precursor”? Please, clarify in the text.

16. Line 322-356 and Fig 7: Not only stem-like T cells are Tcf+, but also naïve and memory cells. The authors state that the difference in tumor growth is due to depletion of stem-like cells. How can one be sure about this? Were naïve DTR+ P14 cells adoptively transferred in the experiments shown in Fig. 7?

17. Line 364/365 (beginning of Discussion): “we showed that the optimal synergy with RT and anti-PD-L1 depends on this TdLN subset”. Isn’t this an overinterpretation because the experiments did not prove that “this TdLN subset” is required? Moreover, I assume the TEFF-STEM subset is meant but no markers are mentioned. Please, specify the markers here.

18. Line 369/70: “A novel differentiation program with more robust stem-like T cell proliferation”. In which figure do the authors show more T cell proliferation?

19. Line 374/375: “our findings suggest that the RT ...is enough, by itself, to promote stem-like T cell expansion and differentiation initiated in the TdLN”. Didn’t the authors only find more stem-like, including more TEFF stem cells in the RT/aPD-L1 combination group?

20. Discussion, line 389-392: these sentences read as if Tpex cells may not be relevant precursors in combined RT/aPD-L1 therapy. The authors should discuss their view in more detail to clarify their interpretation.

Minor points:

21. Lines 55-57 (last sentence of abstract): “these data demonstrate a multistep stimulation of stem-like T cells”. Isn’t there a more specific novelty in the paper that could be pointed out?

22. Line 79: introductory sentences about the RT-induced abscopal effect. Refs 12-14 seem not adequate to reflect the state of the field, they are mainly case reports.

23. Lines 81-82: sentence on possible synergy between RT and single agent checkpoint blockade. Again refs. 15-18 seem inappropriate, since they are clinical studies where hardly any abscopal effect has been observed, and one preclinical study where the combination with dual checkpoint blockade, but not monotherapy, was investigated.
24. Ref. 17 does not highlight the combination with single agent checkpoint blockade.
25. Line 82: Ref 18 and 19 are given as example for encouraging clinical results. However, ref. 15 and 19 are identical.
26. Line 90: it is again stated that in the tumor, stem-like cells differentiate into terminally differentiated effector-like cells.
27. Line 127:
The molecular markers in the lower right inset of Fig. 3A (Subtype/color code) are blurred/difficult to read.
28. Line 147: "TDLN immunodepletion". Please clarify what is meant.
29. Line 210-217: "TSTEM cells", please clarify whether the Tcf7+Fos+ stem-like subset is meant here, or all stem-like cells.
30. Line 211 and elsewhere: ...talk about T stem cells from the tumor-draining LN. Are LNs draining the irradiated tumor meant? Please clarify.
31. Fig. 4G and 4H the letters are too small.
32. Line 360/361 (discussion): "focusing on a Tcf1+ stem-like subset". The markers of the subset should be mentioned to better understand.
33. Line 363/364: "once in the tumor, these cells differentiate into terminal effectors"
What are "terminal effectors" here, exhausted cells?
34. Line 373/74 "to the exclusion" is not comprehensible. Please check this sentence for both grammar and meaning.
35. Line 383 "TEFF-STEM dramatically increase with combination therapy...". the FACS data didn't look so impressive to justify this expression.

Reviewer #3

(Remarks to the Author)

In the manuscript "Radiation and anti-PD-L1 synergize by stimulating a stem-like T cell population in the tumor-draining lymph node" (#NCOMMS-24-09148-T), Shen and colleagues investigated the role of CD8+ PD1+ Tcf1+ stem-like T cells in responses to radiation (RT)+anti-PL1 in an immunocompetent preclinical model of melanoma.

The message of this article is important, novel and within the scope of Nature Communications. I have NO concerns about image manipulation or data fabrication.

However, some of the conclusions are not supported by the dataset, and hence must be appropriately revised (or additional experiments performed in this respect). Alongside, some minor issues emerged that need to be addressed, as detailed here below.

Comments:

- Authors used 10Gy irradiation to combine with anti-PDL1 without providing rationale. This radiation dose is not clinically relevant, which raise the question on whether similar effect can be found using clinically relevant radiation doses.
- Only one model was used. Key findings should be performed in another model resistant to anti-PDL1 as monotherapy.
- Authors showed that blockade of T cells egress using FTY720 prevent the trafficking of CD8+ PD1+ Tcf1+ T cells and assumed that the loss of response if due to the absence of this population of B16 tumors. However, it FTY720 is abrogating the egress of all T cells, thus raising the question on whether stem-like CD8+ T cells is responsible for this effect.
- It is known that CD8+ PD1+ CD69+ Tcf1- are terminally exhausted T cells. Authors should comment on whether stem-like CD8+ T cells might become terminally exhausted in their system.
- Fig. 5 and S5: authors concluded that combination therapy leads to a novel differentiation program in the tumor draining lymph nodes stem-like T cells population with migration, expansion and effector differentiation in the tumor. Can authors provide mechanistic insight on why these cells are becoming effector in the tumor and not terminally exhausted? Is there a cytokine responsible for this phenomenon?
- Fig7.: Authors mention that optimal synergy btw RT and anti-PDL1 is required, which is circling back to the initial question on whether 10Gy RT is the optimal RT regimen in their system. Will they achieve the same data using a fractionated regimen that is more relevant or is it a sequencing issue?
- Can author comment on whether this population will be important for responses of anti-CTLA4 or anti-PD1 in irradiated tumors?

Reviewer #4

(Remarks to the Author)

This study investigates the cellular and molecular mechanisms by which combination radiotherapy and immune checkpoint blockade synergize to control tumor growth. It focuses on mouse models, in particular the B16F10 melanoma line with the model LCMV glycoprotein (GP) antigen to track antigen specific responses and the cognate GP-specific P14 model TCR for adoptive transfer experiments. All experiments include tumors engrafted on opposite flanks of the mouse so that the abscopal effect can be evaluated (whether radiation of one tumor influences the other site). The authors focus on the role of stem-like T cell populations and the special role of tumor draining lymph nodes (TdLN) in their development. The study is technically well done with many extensive controls and creative mouse genetics. There is definitely a tendency towards over interpretation and overly-broad conclusions, but this can largely be addressed by changes in the text. We can say that a very useful working model is proposed concerning induction of a particular stem-like subpopulation in the tumor draining lymph node bearing markers of interferon signaling which then traffics to the tumor where it terminally differentiates and controls tumor growth. This will be useful to guide future studies and opens interesting questions concerning the exact differentiation mechanisms into and out of the "Teff-stem" state.

Major comments

1. All mouse experiments contain two implanted tumors to study abscopal effect. While analyses of tumor 1 (irradiated) and tumor 2 (non-irradiated) are presented, no paired analyses are reported to see how much the effects are quantitatively within each mouse. Unless there is a strong reason not to do so, adding such analyses would be highly informative.
2. Line 175 says "Naïve cells were sequenced separately" but I could find no details in the Methods about how these cells were sorted. I am surprised by the need to isolate them separately as lymph nodes, even in tumor bearing mice, would be expected to have naïve T cells. Can the authors provide more details on how these cells were isolated and why there was a need to do so ?
3. The scRNAseq results in both fig 3 and fig 4 should both be presented per mouse (in the supplemental is fine). Supplemental figure S4B does this but only for one population. In figSFB I don't understand the difference between circle radius and position on the vertical axes (or why 0 is at the top of the vertical axes and positive values are below).
4. Statistical tests to evaluate the differences of cell proportions by treatment conditions in figure 4A,B appear to be missing.
5. Line 247 – the adoptive transfer experiment is good, but rather artificial. Is it not possible to use the Gp33+ tetramer to evaluate whether tumor-antigen specific T cells enter this Teff-stem state ?
6. Line 252-254 and figure 4G,H. Given the dramatic change in cell proportions visible earlier in the figure, I do not understand why such pseudo-bulk analysis was done on all cells by condition. It would be much more informative to look for condition-specific expression profiles cluster by cluster.
7. Figure 4I seems potentially at odds with the general model proposed with the later figures. Specifically, the Teff-stem state appears to be a "dead-end" of differentiation here, not a potential intermediate towards Tex cells. My interpretation of the later model though is that these cells traffic to the tumor where they control the tumor while further differentiating towards something like a Tex state. Can the authors provide another interpretation ?

Minor comments

1. Figure 3B is unclear. Are M1 to M5 different mice ? How does a percentage barplot like this reflect the overlap between two repertoires (ie lymph node vs tumor) ?
2. Line 190 refers to figure 3D-E before line 194 refers to Figure 3C. Panels should be re-labelled so they are referred to in order.
3. To demonstrate the added utility of identifying gene expression modules correlated with pseudotime (lines 213 – 215), can these be quantitatively compared to the different clusters and their marker genes ?
4. Line 304-306 does not appear to be supported by the data. Specifically, the text says "combination therapy led to greater expansion of stem-like P14s in both tumor 1 and tumor 2" but figure 5K clearly shows lower levels of stem-like P14 cells. Maybe the authors intended to refer to fig 5I which shows total P14 cells, not the stem-like subset ?
5. Line 353 – 356. This is an over-statement. It is a reasonable hypothesis that endogenous stem-like and terminally exhausted T cells are the cause of the incomplete suppression of tumor control but this is far from proven. The phrasing should be more cautious/qualified.
6. Line 386 – 388. It is not clear that transitory or intermediate phenotype cells can't have a distinct epigenetic signature.
7. Line 399-400. This is another overstatement. The results presented in this manuscript potentially explain (part of) the clinical results referred to but cannot be expected to "definitively" show why they happened.

Version 1:

Reviewer comments:

Reviewer #2

(Remarks to the Author)

The authors addressed my previous concerns satisfactorily.

Reviewer #3

(Remarks to the Author)

Authors have cleared my concerns.

Reviewer #4

(Remarks to the Author)

The manuscript is greatly improved with additional experiments and clearer, more careful interpretations in the text. The authors have addressed all of my important concerns. I only have a few minor edits to suggest for improved readability.

Minor points.

1. There remain several instances where figure panels are referred to out of order (eg Figure 4C is referred to first on line 225 but Figure 4B is referred to first on line 235). I suggest working on figure layout/labelling so that panels are first referenced in the expected order.
2. Line 292 sqys Tcf7 is enriched in Tstem-2 cells compared to other populations but this does not appear to be the case in figure 5D,E. Perhaps the authors meant to refer to a different gene, such as Sell ?
3. In figure 6A,E the axes for CD314 and LY6A are swapped. I suggest being consistent.

We would like to express our sincere gratitude to the reviewers for their time and thoughtful consideration of our manuscript with a revised title, “*Combination radiation and anti-PD-L1 enhance tumor control via stimulation of a PD-1+ TCF1+ CD8+ T cell population in the tumor-draining lymph node.*” We found the reviewers’ feedback very helpful, and in addressing their comments, we have uncovered additional insights into the effects of radiotherapy (RT) and anti-PD-L1 on the PD-1+ TCF1+ T cell population. These new findings are summarized in the point-by-point rebuttal below. Briefly, for the more salient points:

- 1) We renamed the major cell populations identified by scRNA-seq analysis in the untreated tumor draining lymph node (TdLN) to better reflect the conclusions of our study. Specifically, we renamed T_{STEM} to T_{STEM-1} and T_{EFF-STEM} to T_{STEM-2}, and T_{EX} to Effector-like/terminally differentiated in the updated version. The updated names are more neutral and imply less about functional states as well as accounts for the diversity of phenotypes within the TCF-1- population. These names are now consistent throughout the manuscript.
- 2) We further validated and characterized our scRNA-seq data by flow cytometry. To do this, we sorted the newly defined tumor antigen specific T_{STEM-2} from the TdLN population following RT+ anti-PD-L1 using LY6A and CD314 and the transgenic P14 system. These markers were selected based on their elevated and specific expression in scRNA-seq data for this newly defined subset. We then transferred T_{STEM-2} into a separate cohort of tumor-bearing mice and observed the T_{STEM-2} differentiation into TIM-3+ cells in the tumor confirming their ability to migrate from the TdLN to the tumor and their transitional or intermediate status. The data are included in the entirely new Figure 6.
- 3) We repeated our experiments using 8 Gy x 3 fractions + anti-PD-L1 with the B16F10GP cell line. We obtained similar tumor control at both the primary and the abscopal sites, consistent with the results observed following a single 10 Gy dose. These data are shown in Supplementary Figure 2A-E.

- 4) We extended our experiments to include the YUMM1.7 cell line, which is resistant to anti-PD-L1, to validate our key findings. With RT and anti-PD-L1 treatments, we observed similar tumor controls and increase in the numbers of PD-1+ TCF-1+ cells and TIM-3+ cells, similar to the results seen in the B16F10GP model (Supplementary Figure 2F-J)

The paper is now organized with the following figures:

Figure 1. RT + anti-PD-L1 promote an increase in intra-tumoral PD-1+ TCF-1+ and PD-1+ TIM-3+ CD8+ T cells. In the updated version, we show GZMB and TOX histograms for PD-1+ TCF-1+ cells and TIM-3+ cells.

Figure 2. The TdLN supplies the tumor with PD-1+ TCF-1+ CD8+ T cells following RT + anti-PD-L1. This is unchanged from the previous Figure 2.

Figure 3. RT promotes the expansion and differentiation of TdLN PD-1+ TCF-1+ T cells which is enhanced with anti-PD-L1. This is the same figure as the previous Figure 5 but moved to Figure 3.

Figure 4. ScRNA-seq analysis identified multiple CD8+ PD-1+ TCF-1+ T cell subsets in the TdLN. Previous Figure 3. We renamed the major cell populations identified by scRNA-seq analysis to better reflect the conclusions of our study. Specifically, we identified them as numerical clusters and then renamed T_{STEM} to T_{STEM-1} and $T_{EFF-STEM}$ to T_{STEM-2} , and T_{EX} to T_{TD} in the updated version.

Figure 5. Combination RT + anti-PD-L1 expands a novel TCF-1+ subset in the TdLN. Previous Figure 4. We have now included RNA velocity analysis as part of our revised approach and removed the pseudotime data. And included both statistical analysis and gene expression patterns broken down by the subsets (Panel C, Panel E).

Figure 6. T_{STEM-2} cells differentiate into TIM-3+ T cells in the tumor. This is a new figure. We further characterized and sorted the newly defined T_{STEM-2} population using LY6A and CD314, based on the scRNA-seq data and flow cytometry validation data. We then transferred these cells into a separate cohort of tumor-bearing mice and observed their differentiation into TIM-3+ cells.

Figure 7. PD-1+ TCF-1+ CD8+ T cell depletion attenuates the enhanced tumor control of RT + anti-PD-L1. We replaced Figures 7E and 7F with new data showing Tcf-1+ LY6A+ cell depletion. The figure is otherwise unchanged.

REVIEWER COMMENTS

Reviewer #1 (Remarks to the Author): with expertise in T cell biology, trafficking

In this manuscript, Buchwald and coworkers seek to identify the mechanism that underlies the synergistic effect of radiotherapy (RT) and anti-PD-L1 immunotherapy using B16F10 melanoma cells expressing the LCMV neoantigen gp33. Using a combination of gp33-tetramers, P14 TCR transgenic mice, scRNA sequencing analysis, and a new TCF1-DTR model, the authors demonstrate an increase in two stem-like subsets (TCF1+Fos+ (Tstem) population and a TCF1+Tox+ precursor exhausted (Tpex) population) and a new TCF1+Ly6A+ effector stem (Teff stem) intermediate in the tumor draining LN. In response to radiation therapy, the LN stem-like population does not change in number; however, they appear increased in the tumor. These data are interpreted to conclude that RT is sufficient to promote stem like T cell expansion, which the authors highlight as a novel finding in their study (lines 374-8). But this conclusion should be properly tempered because it does not distinguish between the ability of RT to induce Tstem populations through 1) priming of naive T cells, which is an expected effect of RT vs. 2) expansion of pre-existing Tstem cells, which occurs following PD-1 blockade but has not been reported as a direct effect of RT. The former mechanism seems predictable, while the latter if substantiated with new data would be novel.

We thank the reviewer for the very detailed and insightful comments. The comments have helped demonstrably improve the manuscript.

In Figure 3, we demonstrate that adoptive transfer of antigen specific, PD1+ CD44+ TCF1+ T cells (non-naïve) from TDLN are stimulated to expand and differentiate with RT alone and this effect is further enhanced with combination therapy. While, we certainly agree, this does not exclude an impact of RT on the naïve T cell population, it does demonstrate the impact on pre-existing antigen experienced TCF1+ population.

This study claims to have identified a novel differentiation intermediate, a T effector stem (Teff stem) subset, marked by TCF1 and Ly6A expression that appears in the tumor draining LN but not in the tumor following RT and anti-PD-L1 combined therapies. Teff stem cells were identified by scRNAseq analysis. Their intermediate state was determined by Monocle pseudotime trajectory analysis, which inferred that Tstem cells differentiate into two distinct fates, Tpex and Teff stem. A bifurcated Tstem differentiation pathway and the identification of a new Teff stem intermediate are potentially novel findings. However, pseudotime trajectory analysis is highly biased because the user chooses the root/starting point of the trajectory and inferences are made based on progressive changes in overall gene expression. RNA velocity is a far better trajectory method that is unbiased and does not require user identification of root states. Instead, it uses ratios of spliced and unspliced transcripts as an indication of active and past gene transcription. The authors need to reanalyze their trajectory analysis with RNA velocity to compare differentiation trajectories. The current pseudotime trajectory is also confusing since it predicts that Tex cells only arise from Tpex. Yet, the authors propose that with combo therapy, Teff stem cells in the LN give rise to Tex in the tumor.

Again, a very good point made by the reviewer. We have now removed the somewhat confusing pseudotime trajectory and included a re-analysis using RNA velocity. This new analysis shows the Tex (now renamed Effector-like/TD) are given rise to by the Tpex (Figure 5F) as has been shown previously by multiple groups including Huang et al. Cell 2022. It also shows a pathway from Tstem-1 to Tstem-2; we are now careful to avoid claiming that based on the RNA velocity the Tstem-2 can produce effector-like/TD cells. To determine the ability of tumor antigen

specific TdLN Tstem-2 to further differentiate into effector-like cells in the tumor, we performed adoptive transfer experiments described below (New Figure 6). The ability of the Tstem-2 to traffic to and give rise to TIM-3+ cells in the tumor is validated in Figure 6F-H.

This manuscript proposes the following differentiation pathway: Tstem cells give rise to Teff stem cells in the draining LN in response to combo therapy followed by Teff stem cell migration to the tumor where it expands and differentiates into Tex cells. The fate of the well-established Tpex subset in this progression are not discussed and need to be examined since the field considers Tpex cells in the LN to be the primary population responding to PD-1 blockade. The proposed differentiation pathway is also not adequately linked. If Teff stem cells are trafficking from LN to tumor, then they should be observable in the blood. If Teff stem cells that infiltrate the tumor give rise to Tex cells then there should be a progression from TCF1+LY6A+ through intermediates to TCF1-Tim3+ cells. This should be demonstrated by flow cytometry and by adoptive transfer of intermediates to temporally link downstream states.

This is an excellent point. We agree that most literature considers the TCF1+ TOX+ cells the primary population responding to PD-1 blockade. Of note, Huang et al. Cell 2022 showed that the Tstm (TCF1+ TOXneg) population in the TdLN responds more robustly to PD-1 blockade than the Tpex and, while not directly related, are superior at tumor control following adoptive transfer. Here, we show that tumor-specific Tstem-2 are found in the blood by flow – defined as P14 PD-1+ TCF-1+ LY6A+ CD314+ (Figure 6E). Using an adoptive transfer system with our tumor specific P14 T cells, if we sort P14 PD-1+ CD44+ T cells co-expressing CD314 and LY6A (specific markers based on our single cell RNA-seq data for the Tstem-2) from the TdLN, they are found in the tumor and have the capacity to differentiate into intermediates (TCF1-TIM3-) and effector-like cells (GZMB+ and TIM3+ TCF1-) (Fig 6F-H). We do not wish to state with any certainty however that the Tstem-2 do not also transition through a Tpex like state in the tumor and therefore we do not assert this in the manuscript. We thank the reviewer kindly for helping calibrate our conclusions and language.

The overall therapeutic effect of single and even combination therapy is rather modest. This is perhaps because B16F10 and YUMM1.7 cell lines have relatively low immunogenicity and poor

responsiveness to anti-PD-1 therapy. PD-1 responsive tumors (MC38, YUMMER1.7) should be examined in combination with RT to determine whether the Teff stem intermediate is induced as a general mechanism of RT and PD-1 combination therapy.

We thank the reviewer for the suggestion. We examined the Tstem-eff cells (now renamed Tstem 2) in the YUMMER1.7 model and observed an increase in their population with RT and anti-PD-L1 combination therapy (Figure S6A-C).

Additional specific concerns:

1. The combination treatment appears to have an additive rather than synergistic effect. If authors wish to conclude otherwise, they should mathematically demonstrate synergism.

We have removed the word synergistic from the manuscript to avoid overstating the observations.

2. The scRNA- and TCR-seq data presented in Fig 3 lacks proper explanation in the text, figure legend and methods.

Thank you. In the revised version we have expanded and revised the explanations for former Figure 3, now Figure 4 in the results section with heading *ScRNA-seq analysis identified multiple CD8+ PD-1+ TCF-1+ T cell subsets in the TdLN*. The corresponding methods section and figure legends has also been significantly expanded.

- Fig 3A. Which specific datasets were used to determine subset identities? Was a minimum gene enrichment score cutoff imposed to give the authors confidence that each subset is accurately defined? They should use a non-biased method for cluster identification instead.

Thank you for the comments on subset identification in the new Fig 4A. In response, we have provided additional details in the Methods section to clarify the datasets and approaches used. Specifically, we used publicly available single-cell RNA sequencing datasets with comprehensive CD8+ T cell type annotations, including those from Huang et al., *Cell*. 2022, which also examined CD8+ T cells from the TdLN, and Miller et al., 2019, which employed a

similar mouse model. To ensure unbiased cluster identification, we applied the Leiden algorithm, a graph-based method known for detecting clusters objectively, rather than using a gene enrichment score cutoff. We validated the identified clusters by comparing them to key cell type markers and performing differential expression analysis to confirm their accuracy. These updates are now included in the revised manuscript.

- Fig 3B. What does M1 - M5 mean? Are these sorted cells from non-treated, RT, and/or aPD-L1 treated mice? There is no comparison to a non-tumor draining LN.

The samples labeled M1 - M5 in Figure 4B (formerly Figure 3B) represent sorted CD8+ PD-1+ T cells from individual untreated mice (mouse 1, mouse 2, etc.), collected to investigate the baseline clonal relationship between TdLN-derived CD8+ PD-1+ T cells and CD8+ T cells within the tumor without treatment. This is now described in the figure legend. We recognize the value of comparing these findings to non-tumor draining lymph nodes, however, we show in Supplementary Figure S3A there are limited antigen specific cells in the non-TdLN or the spleen for that matter. Our current analysis focuses on untreated conditions in the TdLN to establish a foundational understanding of T cell populations and their clonal relationships. To clarify, we have revised the manuscript to explicitly state that M1 - M5 represent untreated mice in section *ScRNA-seq analysis identified multiple CD8+ PD-1+ TCF-1+ T cell subsets in the TdLN.*

- Fig 3C. The proportions of all 5 T cell subsets in the tumor and draining lymph node need to be verified by flow cytometry. Addition of gp33 tetramers to identify tumor-specific T cells and preferably also a non-model antigen such as Trp2 should be used to determine whether these subsets represent tumor-specific CD8 T cell populations.

Thank you for your suggestion. We have validated the main subsets using the adoptive p14 T cell transfer system (new Figure 6A-B). The p14 approach was used because it provides more robustly detectable tumor specific T cells numbers in the TDLN for analysis.

- Fig 3F. Monocle 3 pseudotime analysis was conducted to infer single cell trajectories.

However, this analysis is biased and flawed given that the authors selected the TCF^{high} cells in the draining lymph node as the root/starting population for trajectory analysis. Instead, RNA velocity/scVelo analysis, which infers population transitions based on active RNA splicing of actively transcribed genes should be conducted for trajectory inference.

This pseudotime trajectory analysis has been removed from the figure as it is redundant with the RNA velocity performed in Figure 5F.

-Fig 3G/H. Tstem and T exhausted signature density plots are depicted but how they were calculated is not described.

In response to your feedback, we have included a detailed description of the methodology for calculating the stem and exhausted signature density plots in the revised manuscript. Briefly, we utilized the R packages Nebulosa (Alquicira-Hernandez et al., *Bioinformatics*. 2021) and UCell (Andreatta et al., *Computational and Structural Biotech.*, 2021). Gene signatures for stem and exhausted cells were curated from established literature (Andreatta et al., *Nat Commun*. 2021; Huang et al., *Cell*. 2021; Sade-Feldman et al., *Cell*. 2019; and Miller et al., *Nat Immunol*. 2019). We then applied the UCell package to calculate enrichment scores for each gene signature per cell using a rank-based method. Finally, the Nebulosa package was used to generate smoothed density plots via kernel density estimation, allowing for clear visualization of signature score distributions across the cell populations.

3. Fig 4A-B: As above for 3B, the proportions of all 5 T cell subsets in the tumor and draining lymph node need to be verified by flow cytometry with MHC tetramers for all 4 groups if the authors wish to highlight this population as the key subset involved in responsiveness.

These data are included in the new Figure 6. We have also downplayed the ultimate importance of this subset and make more nuanced claims.

- Fig 4D-H: Text and legend are not clear as to whether data are from tumor or tumor draining LN.

Thank you. We have updated the text and legend in the revised version of the manuscript.

- Fig 4F: transfer of 2.5 million P14 TCR tg cells is excessive and unlikely to represent native responses to therapy.

We apologize for the typo; the correct cell number (2.5×10^5) is provided in the Methods section. We have corrected this in the revised version.

- Fig 4I: pseudotime analysis should be replaced with RNA velocity to avoid bias.

Thank you. We have performed RNA velocity analysis in the revised version (Fig 5F). The results are discussed above.

- Fig S4B: y axis does not make sense. Does this represent a p-value or enrichment score? S4D: signature density plots are depicted but how they were calculated is not described.

To improve clarity, we have removed this panel from the revised manuscript. For Supplementary Figure 5 (previously S4D), we have now included a detailed description of the methodology used to calculate the signature density plots in the methods section Lines 616-623, similar to that provided for Figures 4G and 4H. Briefly, we utilized the R packages Nebulosa (Alquicira-Hernandez et al., *Bioinformatics*. 2021) and UCell (Andreatta et al., *Computational and Structural Biotech.*, 2021) to generate these plots, ensuring accurate visualization of the distribution of enrichment scores across cell populations.

4. While generation and use of TCF1-DTR mice to selectively deplete Tstem cells in the LN and

tumor are commendable, it is predictable that TCF1+ cells are required for responsiveness to combined treatment.

We thank the reviewer for acknowledging the use of TCF7-DTR mice in our study. While the importance of PD-1+ TCF1+ cells in various immune settings is established, our findings reveal that TdLN PD-1+ TCF-1+ T cells specifically contribute to the enhanced therapeutic interaction observed with RT + anti-PD-L1 therapy and appear to undergo expansion and differentiation with RT alone (Fig 3I, K). This requirement was not evident prior to our depletion experiments utilizing the TCF7-DTR mouse model.

5. gp33-tetramer staining host cells would serve as a better control in Figs 6 and 7 than the bulk T cell population.

With P14 cell transfer there were very few gp33 host cells, which is why we chose bulk T cell population as a control.

6. Are Fig 2E and 2H the same graph?? The last combination column is different, but the other 7 columns are identical.

Fig 2E and 2H are not the same graph, even though they appear very similar. The reason they appear similar is because the TCF1+ cells represent only a small fraction of all endogenous Gp33+ T cells, with the majority being TIM-3+ cells in the tumor. Therefore, the graphs showing the number of Gp33+ T cells look almost the same as the TIM-3+ graph.

Reviewer #2 (Remarks to the Author): with expertise in cancer radiotherapy/immunotherapy

Yang Shen et al. Radiation and anti-PD-L1 synergize by stimulating a stem-like T cell population in the tumor-draining lymph node.

Shen et al. investigated the mechanism underlying the synergism of radiotherapy and anti-PD-L1 in local and distant (abscopal) tumor control in mouse melanoma models. The study focuses on a CD8+ stem-like subset in the tumor-draining lymph nodes (TDLN). The main claim of the study is a novel differentiation program in TDLN stem-like cells which leads to the expansion of these

cells into effector cells in tumor tissue and which is responsible for the observed synergism of RT and anti-PD-L1. The question is interesting, the methods are modern, and there are interesting observations. However, the conclusions seem to me to be partly over-interpreted and the data and conclusions are not consistently clearly described in the text.

Major points:

1. Line 51 (Abstract): “RT + anti-PD-L1 induces a novel differentiation program in the TDLN population which leads to their expansion and differentiation into effector cells in the tumor”. With the methods used and data presented, can the authors be sure that this is indeed a new differentiation program? As markers for more differentiated non-stem-like cells PD1+TIM3+TCF- were used throughout the paper which are the markers typically used for (effector-like) non-terminally and terminally exhausted, but not normal effector, T cells.

Thank you for your thoughtful feedback regarding our characterization of the T cell subsets. This verbiage has been removed from the abstract and the differentiation program language has been couched in far more cautionary terms and removed from the abstract entirely. We use the term effector-like for subsets which express high levels of GzmB+ in addition to Tim-3 and TCF-1-, otherwise the subsets were exclusively identified by their markers. We have altered our terminology throughout for referring to more generically TIM-3+ cells, as we acknowledge Tim-3+ cells can potentially fall into a transitory and more terminally differentiated subset (Hudson et al. *Immunity* 2019). As the main focus of this study was to evaluate the TCF-1+ subsets in detail, our subgroup analysis was primarily focused on this population, but we appreciate and acknowledge more clearly the importance of the diversity within the TCF-1- population.

2. Line 52: “effector cells”. Do the authors mean exhausted T cells?

We have removed this ambiguous terminology.

3. Line 69-71: describe the subsets of TILs as: “progenitor stem-like CD8+ T cells and terminally differentiated effector like cells (TE)”. Not all non-stem-like cells in the TME are terminally differentiated. Therefore, this should be described better.

This is a very helpful point. We agree as acknowledged in point 1 that the TCF-1- population contains both transitory/effector-like cells and exhausted/terminally differentiated and therefore, we are no longer referring to effectors, effector like or terminally differentiated cells in the TCF-1- population when TIM-3 alone is used as a marker. When limited markers are used, we identify the subsets only by the markers expressed and emphasize the focus of the study on the TCF-1+ population.

4. Line 91: „Huang et al. demonstrated that the TdLN stem-like population is an important mediator of the anti-PD-1/L1 response.” “our earlier work suggested...” The authors cite Huang et al. (ref. 22), but in this reference a differentiation pathway of anti-PD-1/PDL-1-responsive T cells (emphasizing the importance of TSM bona fide memory T cells rather than TPEX cells) is reported which is different from other prominent papers in the field. These different views do not become clear by reading this part of the introduction.

Thank you for your insightful comment. This has been removed from the introduction to avoid confusion, and distract from the main thrust of the intro.

5. Line 118/119: the authors indicate growth of tumor 2 was significantly reduced after RT monotherapy; however, there is no indication of statistical significance between control and RT-treated mice for the abscopal tumor (Fig. 1B).

We thank the reviewer for pointing this out. We have updated the description accordingly in the revised version. We replaced our previous description "Tumor 1 and tumor 2 growth were significantly reduced with RT alone" with " Tumor 1 growth was significantly reduced with RT alone, and tumor 2 growth also exhibited a strong trend towards slowed growth."

6. Line 132-134: here, the markers of the two populations mentioned (stem-like and TE) should be given in the text. According to the figures, the TE = terminally differentiated subset was identified as TCF1-Tim3+. However, the TCF1-Tim3+ population has been described by others to also contain non-terminally differentiated exhausted cells. Throughout the MS, the authors only consider stem-like and terminally exhausted T cells. Why?

Thank you. We agree and have noted above that we did not mean to imply all the TCF-1- cells were terminally exhausted and acknowledge the presence of a transitory/effector-like subset. However, as this manuscript and the studies herein are focused on the TdLN and the TCF-1+ T cells, we have changed the language and refer to TIM-3+ cells by their marker of expression rather than potential functional characteristics.

7. Line 151: again, it is stated that most of the tumor-specific TILs in the tumor are terminally differentiated without measuring non-terminally differentiated cells (which are also included in the TCF-TIM3+ population).

Please see our response above.

8. Line 151 and further, and Fig. 2 A, B and further down: here and in other figures, corresponding legends and throughout the text it is unclear in which tumor (primary or secondary) the T cells were analyzed.

Thank you. We have clarified this throughout in the text.

9. Line 178-181: The authors state they found populations with signatures similar to ref 22 (Huang et al.). Line 186-190: “Both previously described...subtypes and new groups”. It is not fully clear which subtype was previously known. Do the authors refer to the Tcf7+ Fos+ as previously known? This should be clarified.

Thank you for pointing out this ambiguous phrasing, it has been largely removed. In the revised manuscript, we have specified which CD8⁺ PD-1⁺ T cell subsets were previously known and which were newly identified. Specifically, we have clarified that both the Tcf7⁺ Fos⁺ T_{STEM-1} subset and the Tcf7⁺ Tox⁺ T_{PEX} subset were previously described in the literature (reference 22, Huang et al.), while the non-canonical Tcf7⁺ Ly6a⁺ T_{STEM-2} subset is newly identified. Lines 214-223.

10. Line 190: Tcf7 and GzmB are given as typical markers for the T_{EFF-STEM} population. Why was GzmB chosen? In Fig. 3E, other markers than GzmB are much more strongly expressed in this population.

We have updated our analysis to use *Ly6a* and *Klrl1* as primary markers for the T_{STEM-2} subset (previously T_{EFF-STEM}), based on their stronger expression and flow cytometry validation. This adjustment provides a clearer identification of the T_{STEM-2} subset in our study and we have renamed the subset to avoid a bias towards a specific function.

11. Fig. 2, 3, and 4: cells from TdLNs were analyzed. Were these lymph nodes draining the irradiated or the non-irradiated tumor?

These lymph nodes were draining the irradiated tumors. This has been made clear in the text in Lines 165-166.

12. Line 235-244: Why was Ly6a chosen and what type of molecule is it? By FACS (Fig. 4D, F) the Ly6a⁺ population seems to be much smaller than in the RNAseq data. Please explain.

We chose Ly6a (Sca-1) due to its established role as a marker of stem and progenitor cells (Ito et al., Blood, 2003); it also expressed by memory T cells (Whitmire et al. EJI 2010) with significant expression in what is now referred to as T_{STEM-2} (previously T_{EFF-STEM}) relative to other CD8⁺ PD1⁺ T cell subtypes. The differences between FACS and scRNA-seq can reflect inherent technical distinctions between protein and mRNA-based detection. While it is uncommon for protein and transcript levels to align perfectly (Gry et al., 2009; Liu et al., 2016), we selected CD314 and Ly6a as markers that were notably differentially expressed in the transcriptional data,

facilitating clear identification of the T_{STEM-2} population by flow. Importantly, in figure 6 an expected, based on the single cell data, percentage of the TdLN population is LY6A+.

- Gry, M., R. Rimini, S. Strömberg, A. Asplund, F. Pontén, M. Uhlén and P. Nilsson (2009). "Correlations between RNA and protein expression profiles in 23 human cell lines." *BMC Genomics* **10**(1): 365.
- Liu, Y., A. Beyer and R. Aebersold (2016). "On the Dependency of Cellular Protein Levels on mRNA Abundance." *Cell* **165**(3): 535-550.

13. Fig 4E and line 244-246 states that the T_{eff}/stem population has less Tcf than T_{STEM} cells but more than T_{EX}. Do T_{EX} cells express Tcf?

We apologize for the confusion. We have updated the manuscript to clarify that T_{EX} (Effector-like/TD) cells do not express *Tcf7*, with only a very small fraction of cells showing minimal *Tcf7* expression, consistent with their definition as TCF-1-negative. This distinction aligns with our criteria for defining the Effector-like/TD subset. The revised clarification can be found on Line 221-222.

14. Fig. 4I and line 273-276: Fig. 4I suggests that T_{STEM} cells are more immature than T_{PEX} and T_{EFF-STEM} cells. In Fig. 3, T_{STEM} cells represent the TCF+ Fos+ population. Is the T_{STEM} population in Fig. 4I the same as the T_{STEM} population in Fig. 3A or is T_{STEM} here used as a general term for stem-like cells?

Throughout the revised text and figures, we now specifically refer to the T_{STEM} population as T_{STEM-1}. This adjustment aligns with our depiction of the Tcf7+ Fos+ population originally shown in Figure 3A (now updated to Figure 4A). Additionally, throughout the rest of the manuscript we removed the term stem-like cells and simply call them PD-1+ TCF-1+ cells as this is a pool of phenotypes by flow which include the Tstem-1, Tstem-2, and T_{pex}.

15. Line 297-98: what do you mean by “the TdLN stem-like precursor”? Please, clarify in the text.

We apologize for the confusion. This phrasing has been removed as it causes confusion in the restructuring of the revised manuscript.

16. Line 322-356 and Fig 7: Not only stem-like T cells are Tcf+, but also naïve and memory cells. The authors state that the difference in tumor growth is due to depletion of stem-like cells. How can one be sure about this? Were naïve DTR+ P14 cells adoptively transferred in the experiments shown in Fig. 7?

We thank the reviewer for their comments. We acknowledge that Tcf 7 expression is not exclusive to stem-like T cells but also occurs in naïve and memory T cells. However, in this experiment at the time of depletion and afterwards (for the DTR- cells), the TCF1+ T cells also express PD-1 (as is now shown in updated Supplementary Figure 7B-C and Figure 7C, 7I).

17. Line 364/365 (beginning of Discussion): “we showed that the optimal synergy with RT and anti-PD-L1 depends on this TdLN subset”. Isn’t this an overinterpretation because the experiments did not prove that “this TdLN subset” is required? Moreover, I assume the TEFF-STEM subset is meant but no markers are mentioned. Please, specify the markers here.

We agree that this is an overstatement and it has been removed. We were referring generally to the PD-1+ TCF-1+ cells.

18. Line 369/70: “A novel differentiation program with more robust stem-like T cell proliferation”. In which figure do the authors show more T cell proliferation?

Thank you for pointing this out. We have removed the word proliferation. What was meant was an increase in numbers.

19. Line 374/375: “our findings suggest that the RT ...is enough, by itself, to promote stem-like T cell expansion and differentiation initiated in the TdLN”. Didn’t the authors only find more stem-like, including more TEFF stem cells in the RT/aPD-L1 combination group?

In Figures 3H–3L, we show that RT alone increased the number of P14s in the tumor and also induced the differentiation of the PD-1+ TCF1+ T cells into TIM3+ TCF1- cells. But, we agree that the combination groups is far superior at increasing the overall TCF-1+ cells and then STEM-2 subset. We did, however, wish to emphasize that RT alone can promote PD-1+ TCF-1+ T cell differentiation which is a novel observation.

20. Discussion, line 389-392: these sentences read as if T_{pex} cells may not be relevant precursors in combined RT/aPD-L1 therapy. The authors should discuss their view in more detail to clarify their interpretation.

In the discussion, we have clarified this point. In our study, we observed this T_{STEM-2} subset in the TdLN and showed through adoptive transfer experiments that T_{STEM-2} cells migrate to the tumor, where they complete their differentiation program into TIM-3+ GZMB+ cells. Although our study suggests that the T_{PEX} subset may be bypassed in the TdLN following the RT + anti-PD-L1, the T_{STEM-2} may undergo transient T_{PEX} differentiation in the tumor prior to TCF-1 downregulation²². Future studies will evaluate this in more detail and we do not draw any firm conclusions in the results shown here. Future studies will also help determine whether the T_{STEM-2} subset may serve as superior precursor for adoptive cell therapy and can generate TIM-3+ GZMB+ with more potent effector potential. Lines 439-495

Minor points:

21. Lines 55-57 (last sentence of abstract): “these data demonstrate a multistep stimulation of stem-like T cells”. Isn’t there a more specific novelty in the paper that could be pointed out?

The abstract has been rewritten to more succinctly summarize the main findings of the manuscript and synthesize the translational potential.

22. Line 79: introductory sentences about the RT-induced abscopal effect. Refs 12-14 seem not adequate to reflect the state of the field, they are mainly case reports.

References have been updated with more up-to-date literature on the abscopal effect. PMID: 34681719, 29449659, and the seminal paper, 25754329

23. Lines 81-82: sentence on possible synergy between RT and single agent checkpoint blockade. Again refs. 15-18 seem inappropriate, since they are clinical studies where hardly any abscopal effect has been observed, and one preclinical study where the combination with dual checkpoint blockade, but not monotherapy, was investigated.

The sentence has been restructured and we now included the additional following citations which feature some data looking at RT + single agent checkpoint. Line 87.

1. Hammerich, L.; Marron, T.U.; Upadhyay, R.; Svensson-Arvelund, J.; Dhainaut, M.; Hussein, S.; Zhan, Y.; Ostrowski, D.; Yellin, M.; Marsh, H.; et al. Systemic clinical tumor regressions and potentiation of PD1 blockade with in situ vaccination. *Nat. Med.* 2019, 25, 814–824.
2. Dewan, M.Z.; Galloway, A.E.; Kawashima, N.; Dewyngaert, J.K.; Babb, J.S.; Formenti, S.C.; Demaria, S. Fractionated but not single-dose radiotherapy induces an immune-mediated abscopal effect when combined with anti-CTLA-4 antibody. *Clin. Cancer Res.* 2009, 15, 5379–5388.
3. Park, S.S.; Dong, H.; Liu, X.; Harrington, S.M.; Krco, C.J.; Grams, M.P.; Mansfield, A.S.; Furutani, K.M.; Olivier, K.R.; Kwon, E.D. PD-1 restrains radiotherapy-induced abscopal effect. *Cancer Immunol. Res.* 2015, 3, 610–619.

24. Ref. 17 does not highlight the combination with single agent checkpoint blockade.

Thank you, please see response above.

25. Line 82: Ref 18 and 19 are given as example for encouraging clinical results. However, ref. 15 and 19 are identical.

We apologize for the duplication. We have removed the identical reference in the revised version.

26. Line 90: it is again stated that in the tumor, stem-like cells differentiate into terminally differentiated effector-like cells.

Thank you. We have clarified this point to align with the edited nomenclature as noted above.

27. Line 127:

The molecular markers in the lower right inset of Fig. 3A (Subtype/color code) are blurred/difficult to read.

Thank you. We have fixed it in the revised version.

28. Line 147: “TDLN immunodepletion”. Please clarify what is meant.

We apologize for the confusion. TDLN immunodepletion means TDLN irradiation.

29. Line 210-217: “TSTEM cells”, please clarify whether the Tcf7+Fos+ stem-like subset is meant here, or all stem-like cells.

Yes, T_{STEM-1} cell is the Tcf7+Fos+ stem-like subset. Has been clarified in the revised version.

30. Line 211 and elsewhere: ...talk about T stem cells from the tumor-draining LN. Are LNs draining the irradiated tumor meant? Please clarify.

These lymph nodes were draining the irradiated tumors. We have clarified in the text.

31. Fig. 4G and 4H the letters are too small.

Thank you, we have relabeled the letters.

32. Line 360/361 (discussion): “focusing on a Tcf1+ stem-like subset”. The markers of the subset should be mentioned to better understand.

This has been edited in the discussion.

33. Line 363/364: “once in the tumor, these cells differentiate into terminal effectors”
What are “terminal effectors” here, exhausted cells?

This has been clarified in the text and we clarified this ambiguous terminology throughout. We were imprecise and did not clearly articulate differences between effector-like cells and terminally differentiated/exhausted cells.

34. Line 373/74 “to the exclusion” is not comprehensible. Please check this sentence for both grammar and meaning.

We agree with the point raised by the reviewer. We have changed it to “Importantly, prior studies primarily focused on the intra-tumoral T cells, neglecting the T cell subsets present in secondary lymphoid organs” in the revised version.

35. Line 383 “TEFF-STEM dramatically increase with combination therapy...”. the FACS data didn’t look so impressive to justify this expression.

The overstatement has been removed. We used LY6A and CD314 as new markers for TSTEM-EFF (Tstem-2) based on a deeper analysis of our RNA seq results and repeated the experiments with P14 tg tumor specific T cells in the entirely new Figure 6B showing a >10x increase by flow. Additionally, we demonstrate that transfer of this renamed Tstem-2 subset leads to the

subset migration to the tumor and differentiation (Figure 6F-H).

Reviewer #3 (Remarks to the Author): with expertise in cancer radiotherapy/immunotherapy

In the manuscript “Radiation and anti-PD-L1 synergize by stimulating a stem-like T cell population in the tumor-draining lymph node” (#NCOMMS-24-09148-T), Shen and colleagues investigated the role of CD8+ PD1+ Tcf1+ stem-like T cells in responses to radiation (RT)+anti-PL1 in an immunocompetent preclinical model of melanoma.

The message of this article is important, novel and within the scope of Nature Communications. I have NO concerns about image manipulation or data fabrication.

However, some of the conclusions are not supported by the dataset, and hence must be appropriately revised (or additional experiments performed in this respect). Alongside, some minor issues emerged that need to be addressed, as detailed here below.

Comments:

- Authors used 10Gy irradiation to combine with anti-PDL1 without providing rationale. This radiation dose is not clinically relevant, which raise the question on whether similar effect can be found using clinically relevant radiation doses.

We appreciate the point raised by the reviewer. Experiments using 8 Gy X 3 fractions were performed and we obtained similar results to 10 Gy x 1 fraction (Supplemental Figure S2A-E). The data has been added to the revised version of the manuscript.

- Only one model was used. Key findings should be performed in another model resistant to anti-PDL1 as monotherapy.

Experiments using YUMM1.7 cells, which are resistant to anti-PDL1 therapy were performed (Supplemental Figure 2F-J). The data has been incorporated into the revised version of the manuscript.

- Authors showed that blockade of T cells egress using FTY720 prevent the trafficking of CD8+ PD1+ Tcf1+ T cells and assumed that the loss of response is due to the absence of this population of B16 tumors. However, if FTY720 is abrogating the egress of all T cells, thus raising the question on whether stem-like CD8+ T cells is responsible for this effect.

We agree with the point raised by the reviewer. The primary aim of the FTY720 experiment is to demonstrate that the increase in stem-like T cells within the tumor following RT + anti-PD-L1 depends on their egress from the TdLN. By specific depletion of the TCF1+ PD1+ cells in Figures 7, we showed that tumor control induced by the combination RT and anti-PD-L1 was significantly reduced demonstrating specific dependence on the TCF1+ PD1+ cells.

- It is known that CD8+ PD1+ CD69+ Tcf1- are terminally exhausted T cells. Authors should comment on whether stem-like CD8+ T cells might become terminally exhausted in their system.

In our study, we showed that PD-1+ TCF-1+ T cells differentiated into TIM-3+ cells in the tumor, which were predominantly GZMB+ as well (effector-like). However, in the tumor there is more clearly an exhaustion signature in addition to the effector-like. More specifically, in the single cell data in Figure 4 and lines 219, we have now been more clear about the fact that the TIM-3+ TCF-1- population is heterogeneous and contains both effector-like cells and terminally differentiated (exhausted cells).

- Fig. 5 and S5: authors concluded that combination therapy leads to a novel differentiation program in the tumor draining lymph nodes stem-like T cells population with migration, expansion and effector differentiation in the tumor. Can authors provide mechanistic insight on why these cells are becoming effector in the tumor and not terminally exhausted? Is there a cytokine responsible for this phenomenon?

Thank you. We have somewhat de-emphasized this point in response to comments from Reviewers 1 and 2. However, we hypothesize that several factors could contribute to this observed differentiation. For instance, type I interferons are known to promote T cell activation,

migration, and differentiation into effector cells. Citations from Dr. David Brooks' group are included (#35, #36).

- Fig7.: Authors mention that optimal synergy btw RT and anti-PDL1 is required, which is circling back to the initial question on whether 10Gy RT is the optimal RT regimen in their system. Will they achieve the same data using a fractionated regimen that is more relevant or is it a sequencing issue?

Thank you. As we mentioned above, we repeated experiments using 8 Gy X 3 irradiation and we obtained similar results.

- Can author comment on whether this population will be important for responses of anti-CTLA4 or anti-PD1 in irradiated tumors?

A recent study evaluating the impact of anti-CTLA-4 on PD-1+ TCF-1+ T cells is included and discussed in the discussion section. (Wang et al. 2024) lines 455-457.

Reviewer #4 (Remarks to the Author): with expertise in cancer immunology, computational biology

This study investigates the cellular and molecular mechanisms by which combination radiotherapy and immune checkpoint blockade synergize to control tumor growth. It focuses on mouse models, in particular the B16F10 melanoma line with the model LCMV glycoprotein (GP) antigen to track antigen specific responses and the cognate GP-specific P14 model TCR for adoptive transfer experiments. All experiments include tumors engrafted on opposite flanks of the mouse so that the abscopal effect can be evaluated (whether radiation of one tumor influences the other site). The authors focus on the role of stem-like T cell populations and the special role of tumor draining lymph nodes (TdLN) in their development. The study is

technically well done with many extensive controls and creative mouse genetics. There is definitely a tendency towards over interpretation and overly-broad conclusions, but this can largely be addressed by changes in the text. We can say that a very useful working model is proposed concerning induction of a particular stem-like subpopulation in the tumor draining lymph node bearing markers of interferon signaling which then traffics to the tumor where it terminally differentiates and controls tumor growth. This will be useful to guide future studies and opens interesting questions concerning the exact differentiation mechanisms into and out of the “Teff-stem” state.

Major comments

1. All mouse experiments contain two implanted tumors to study abscopal effect. While analyses of tumor 1 (irradiated) and tumor 2 (non-irradiated) are presented, no paired analyses are reported to see how much the effects are quantitatively within each mouse. Unless there is a strong reason not to do so, adding such analyses would be highly informative.

Thank you for your suggestions. We performed a paired analysis of tumor size on day 20 with different treatments, please see the graphs. There was a consistent reduction in both tumors for a given mouse with bilateral reductions for the combination group.

2. Line 175 says “Naïve cells were sequenced separately” but I could find no details in the Methods about how these cells were sorted. I am surprised by the need to isolate them separately as lymph nodes, even in tumor bearing mice, would be expected to have naïve T cells. Can the authors provide more details on how these cells were isolated and why there was a need to do so ?

CD8+CD44-CD62L+ naïve T cells were sorted from lymph nodes of tumor-bearing untreated mice pooled and sequence. We phrased this poorly and confusingly in the original manuscript. We simply meant they were sorted into a separate tube during the original cell sort. Lines 553-554.

3. The scRNAseq results in both fig 3 and fig 4 should both be presented per mouse (in the supplemental is fine). Supplemental figure S4B does this but only for one population. In figSFB I don't understand the difference between circle radius and position on the vertical axes (or why 0 is at the top of the vertical axes and positive values are below).

We have included the scRNA-seq results per mouse (Supplementary Figure 5B. We have removed the confusing figure the reviewer commented on.

4. Statistical tests to evaluate the differences of cell proportions by treatment conditions in figure 4A,B appear to be missing.

We have now included appropriate statistical analyses to address this (Figure 5C).

5. Line 247 – the adoptive transfer experiment is good, but rather artificial. Is it not possible to use the Gp33+ tetramer to evaluate whether tumor-antigen specific T cells enter this Teff-stem state ?

Thank you for your thoughtful suggestion. The TSTEM-EFF cells represent a relatively rare population in the TdLN (now called T_{STEM-2}). To have a substantial number of Gp33+ T cells to better define and evaluate this population, we conducted the adoptive transfer experiment. This was repeated in a more extensive and detailed manner in the new Figure 6. It is difficult with endogenous tetramer+ alone to subclassify the TCF-1+ population due to relatively lower cell numbers. Huang et al. Cell 2022 used a similar approach to us.

6. Line 252-254 and figure 4G,H. Given the dramatic change in cell proportions visible earlier in the figure, I do not understand why such pseudo-bulk analysis was done on all cells by condition. It would be much more informative to look for condition-specific expression profiles cluster by cluster.

We now performed condition-specific expression profiling for each cluster. In Fig. 5E, we now present the individual cluster analyses, illustrating the distinct expression signatures characteristic of each condition.

7. Figure 4I seems potentially at odds with the general model proposed with the later figures. Specifically, the Teff-stem state appears to be a “dead-end” of differentiation here, not a potential intermediate towards Tex cells. My interpretation of the later model though is that these cells traffic to the tumor where they control the tumor while further differentiating towards something like a Tex state. Can the authors provide another interpretation ?

We have removed the somewhat confusing pseudotime analysis, and performed RNA velocity analysis (Figure 5F). This analysis shows T_{STEM-2} (formerly $T_{EFF-STEM}$) may not be terminal “dead-end” state but rather an intermediate state with the potential to differentiate further, however, the trajectory analysis is not convincingly conclusive. We, therefore, have de-emphasized any conclusions from this in the text and simply proceeded to validate these findings with the adoptive transfer experiments in Figure 6F-H which show that sorted T_{STEM-2} have the capacity to differentiate into effector-like cells in the tumor.

Minor comments

1. Figure 3B is unclear. Are M1 to M5 different mice ? How does a percentage barplot like this reflect the overlap between two repertoires (ie lymph node vs tumor) ?

We apologize for the confusion. Yes, M1 to M5 represent five different mice (mouse 1, mouse 2, etc...). In this figure, we detected percentage of unique TCR in the tumor which overlap with the TdLN TCRs. The dark plots highlight unique TCR present in tumors, while the light plots show

TCRs that overlap with those from lymph nodes. We have updated the plots with new ones in the revised version.

2. Line 190 refers to figure 3D-E before line 194 refers to Figure 3C. Panels should be re-labelled so they are referred to in order.

We relabeled the figure in the revised version.

3. To demonstrate the added utility of identifying gene expression modules correlated with pseudotime (lines 213 – 215), can these be quantitatively compared to the different clusters and their marker genes ?

The pseudotime analysis has been removed throughout the manuscript as suggested by other reviewers.

4. Line 304-306 does not appear to be supported by the data. Specifically, the text says “combination therapy led to greater expansion of stem-like P14s in both tumor 1 and tumor 2” but figure 5K clearly shows lower levels of stem-like P14 cells. Maybe the authors intended to refer to fig 5I which shows total P14 cells, not the stem-like subset ?

We thank the reviewer for pointing this out. We should have stated that "combination therapy led to greater expansion of P14s in both tumor 1 and tumor 2 than monotherapy with enhanced differentiation of PD-1+ TCF-1+ T cells into TIM-3+ compared to RT alone". We have updated this in the revised manuscript.

5. Line 353 – 356. This is an over-statement. It is a reasonable hypothesis that endogenous stem-like and terminally exhausted T cells are the cause of the incomplete suppression of tumor control but this is far from proven. The phrasing should be more cautious/qualified.

We agree and we have addressed it in the revised version.

6. Line 386 – 388. It is not clear that transitory or intermediate phenotype cells can't have a distinct epigenetic signature.

We agree and the text has been revised accordingly. Line 439.

7. Line 399-400. This is another overstatement. The results presented in this manuscript potentially explain (part of) the clinical results referred to but cannot be expected to “definitively” show why they happened.

That statement has been removed.

REVIEWERS' COMMENTS

Reviewer #2 (Remarks to the Author):

The authors addressed my previous concerns satisfactorily.

Thank you.

Reviewer #3 (Remarks to the Author):

Authors have cleared my concerns.

Thank you.

Reviewer #4 (Remarks to the Author):

The manuscript is greatly improved with additional experiments and clearer, more careful interpretations in the text. The authors have addressed all of my important concerns. I only have a few minor edits to suggest for improved readability.

Minor points.

1. There remain several instances where figure panels are referred to out of order (eg Figure 4C is referred to first on line 225 but Figure 4B is referred to first on line 235). I suggest working on figure layout/labelling so that panels are first referenced in the expected order.

Thank you for pointing this out. We have adjusted the order of the panels.

2. Line 292 says Tcf7 is enriched in Tstem-2 cells compared to other populations but this does not appear to be the case in figure 5D,E. Perhaps the authors meant to refer to a different gene, such as Sell?

We have removed the description from the text.

3. In figure 6A,E the axes for CD314 and LY6A are swapped. I suggest being consistent.

Thank you for the suggestion. We have now made the axes consistent.